# COMPOSITIONAL DIFFUSION WITH GUIDED SEARCH FOR LONG-HORIZON PLANNING

**Utkarsh A. Mishra, David He, Yongxin Chen and Danfei Xu**
Georgia Institute of Technology
`utkarshm.robo@gatech.edu`

## ABSTRACT

Generative models have emerged as powerful tools for planning, with compositional approaches offering particular promise for modeling long-horizon task distributions by composing together local, modular generative models. This compositional paradigm spans diverse domains, from multi-step manipulation planning to panoramic image synthesis to long video generation. However, compositional generative models face a critical challenge: when local distributions are multimodal, existing composition methods average incompatible modes, producing plans that are neither locally feasible nor globally coherent. We propose Compositional Diffusion with Guided Search (CDGS), which addresses this *mode averaging* problem by embedding search directly within the diffusion denoising process. Our method explores diverse combinations of local modes through population-based sampling, enforces global consistency through iterative resampling between overlapping segments, and prunes infeasible candidates using likelihood-based filtering. CDGS matches oracle performance on seven robot manipulation tasks, outperforming baselines that lack compositionality or require long-horizon training data. The approach generalizes across domains, enabling coherent text-guided panoramic images and long videos through effective local-to-global message passing. More details: `https://cdgsearch.github.io/`

## 1 INTRODUCTION

Synthesizing coherent long sequences is a crucial and challenging task, requiring reasoning over extended horizons. This task arises naturally in various domains: robotic actions must enable future steps, parts of a panorama must align semantically, and subjects in a video must remain consistent across hundreds of frames.

Recent work leverages generative models to learn long sequence distributions [25, 3], with diffusion models [53, 21] gaining popularity for modeling multi-modal data [9, 20]. However, full-sequence data is expensive to acquire, and monolithic models fail to generalize beyond training horizons [10]. As an alternative, compositional generation effectively combines short-horizon local distributions to sample long-horizon global plans [66, 44, 40]—e.g., chaining skills for task planning, connecting images into panoramas, or stitching clips into videos. While this improves data-efficiency and allows extrapolation beyond training data, it intro-

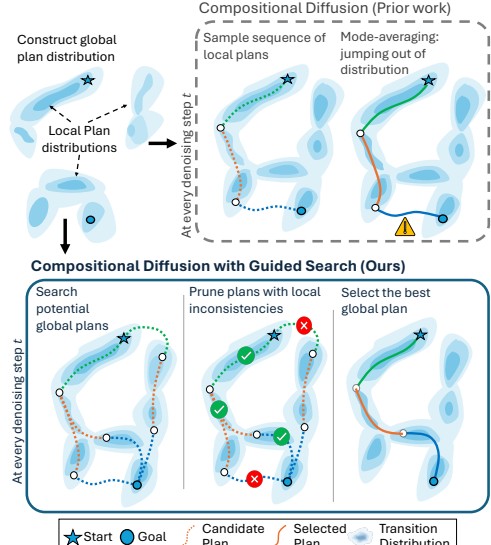

Figure 1: **Compositional Diffusion with Guided Search (CDGS)** composes short-horizon plan distributions to sample long-horizon goal-directed plans directly at inference. Unlike naïve compositional sampling, it explores diverse plans and filters locally inconsistent paths to avoid "mode averaging", yielding globally coherent plans.

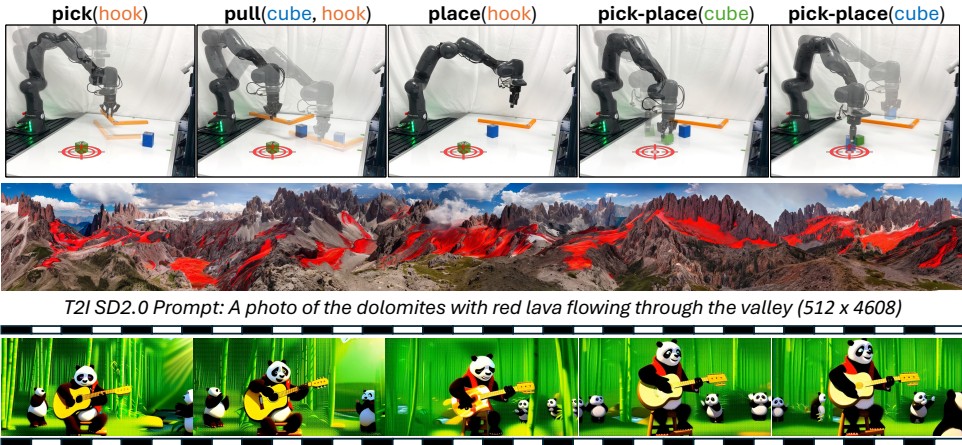

Figure 2: **Applications of CDGS.** (Top) Long horizon motion planning: CDGS discovers a valid multi-step plan to move the blue cube to the green cube's original position via : (1) using the hook to pull blue cube in workspace, (2) displace the green cube to make space and (3) moving the blue cube to the target position. (Mid) CDGS generates coherent panoramic images. (Bottom) CDGS can stitch short clips to generate consistent, longer videos.

duces a **critical challenge**: as local plan distributions become highly *multimodal*, the distribution of global plans inherits combinatorial multi-modality. For example, in the robotics scenario in Fig. 2, because the robot has a large combination of actions and objects it can act on, the search space of possible plans grows exponentially with the length of the planning horizon.

Existing methods for compositional generation offer a promising approach, using *score-averaging* to compose modes of local distributions into a global distribution [66, 44]. However, these methods have an **important limitation**: their inability to handle the combinatorial multi-modality leads them to *average* incompatible local modes (*mode-averaging*), ultimately producing invalid global plans. Addressing such complex multi-modal distributions requires inference methods that jointly reason about compatibility between local modes and effectively navigate the exponentially large search space.

To address the challenge and overcome the limitation, we aim to identify compatible sequences of local modes that compose into a globally coherent plan. Given the diversity and multi-modality of the search space, we take inspiration from classical search techniques and introduce Compositional Diffusion with Guided Search (CDGS), a guided search mechanism integrated into the diffusion denoising process as illustrated in Fig. 1. To facilitate the search during inference, at each diffusion timestep, our method introduces two key components: (i) **iterative resampling** to enhance local-global message passing in compositional diffusion to propose globally plausible candidates, and (ii) **likelihood-based pruning** to remove incoherent candidates that fall into low-likelihood regions due to mode-averaging. Together, these components enable CDGS to efficiently sample coherent long-horizon plans. For robotics tasks, our method outperforms or is on par with baselines that lack compositionality or use long-horizon data for training, respectively. We also show the efficacy of our method in long text-to-image and text-to-video tasks (Fig. 2), producing more coherent and consistent generations.

## 2 BACKGROUND

**Problem formulation.** A long-horizon plan generation problem is characterized by the task of constructing a global plan $\tau = (x_1, \ldots, x_N)$ as a sequence of variables $x_i$, by sampling from the joint distribution $p(\tau)$. The problem becomes goal-directed if $\tau$ must connect a given start $x_1 = x_s$ to a desired goal $x_N = x_g$. Such problems arise in diverse domains: long-horizon manipulation planning, panoramic images, and long videos. While modeling the full joint distribution $p(\tau)$ would directly model all dependencies between any $x_i$, it usually entails end-to-end learning from long-horizon data, which can be infeasible or expensive to obtain. In the absence of long-horizon data, a promis-

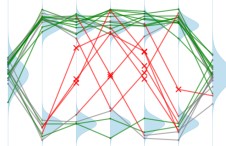 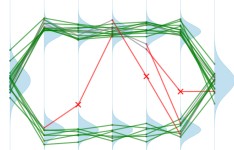 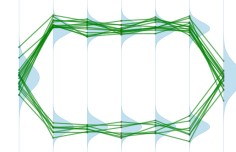

$y_1 \quad y_2 \quad y_3 \quad y_4 \quad y_5 \quad y_6$

$x_1 \ x_2 \ x_3 \ x_4 \ x_5 \ x_6 \ x_7$    $x_1 \ x_2 \ x_3 \ x_4 \ x_5 \ x_6 \ x_7$    $x_1 \ x_2 \ x_3 \ x_4 \ x_5 \ x_6 \ x_7$    $x_1 \ x_2 \ x_3 \ x_4 \ x_5 \ x_6 \ x_7$

(a) the planning domain   (b) naïve composition    (c) with resampling   (d) with resampling + pruning

Figure 3: **Running example.** (a) Consider a 1D-domain of $\{x_{1:7}\}$ variable distributions and $\{y_{1:6}\}$ feasible directed transitions between the variables. There are two feasible long-horizon plans from start ($x_1$) to goal ($x_7$): one through the top and one through the bottom. (b) in naive-composition, sampled plans may choose to start in the top and end at the bottom, or vice versa. When this happens, the intermediate models $\{y_{2:5}\}$ will average the modes of intermediate variables $\{x_{2:6}\}$ to satisfy both constraints, manifesting in infeasible transitions (red) (c) adding **iterative resampling** reduces the frequency of mode-averaging (d) adding **pruning** eliminates plans with infeasible $y$

ing strategy is to approximate the joint distribution $p(\tau)$ with a factor graph of overlapping local distributions that can be learned from short-horizon data. For the joint variable $\tau = (x_1, x_2, \ldots, x_N)$, we construct a factor graph [30] connecting variable nodes $\{x_i\}_{i=1}^N$ and factor nodes $\{y_j\}_{j=1}^M$, where each factor $y_j$ represents the joint distribution of contiguous subsequences of $\tau$. For example, we represent $\tau = (x_1, x_2, \ldots, x_5)$ with factors $y_1 = (x_1, x_2, x_3), y_2 = (x_3, x_4, x_5)$. With this, we construct the joint distribution $p(\tau)$ using the Bethe approximation [64]:

$$p(\tau) := \frac{p(x_1, x_2, x_3)p(x_3, x_4, x_5)\ldots}{p(x_3)\ldots} = \frac{\prod_{j=1}^M p(y_j)}{\prod_{i=1}^N p(x_i)^{d_i - 1}} \tag{1}$$

where $d_i$ is the degree of the variable node $x_i$. This representation enables sampling from the long-horizon distribution $p(\tau)$ using only samples drawn from a short-horizon distribution $p(y)$.

**Diffusion models.** Diffusion models are defined by a forward process that progressively injects noise into the data distribution $p(y^{(0)})$ and a reverse diffusion process that iteratively removes the noise by approximating $\nabla \log p$ to recover the original data distribution. For a given noise injection schedule $\alpha_t$, forward noising adds a Gaussian noise $\varepsilon$ to clean samples s.t. $y^{(t)} = \sqrt{\alpha_t}y^{(0)} + \sqrt{1 - \alpha_t}\varepsilon$. With $p_t$ being the distribution of noisy samples, the denoising is performed using the score function $\nabla_{y^{(t)}} \log p_t(y^{(t)})$ often estimated by a neural network $\varepsilon_\theta(y^{(t)}, t)$ learned via minimizing the score matching loss [23] given by $\mathbb{E}_{t, y^{(0)}}[|\varepsilon - \varepsilon_\theta(y^{(t)}, t)|^2]$. Such a score function allows denoising the noise samples via sampling from

$$p(y^{(t-1)}|y^{(t)}) = \mathcal{N}\left(y^{(t-1)}; \sqrt{\alpha_{t-1}}\hat{y}_0^{(t)} + \sqrt{1 - \alpha_{t-1} - \sigma_t^2}\varepsilon(y^{(t)}, t), \sigma_t^2 \mathbf{I}\right) \tag{2}$$

where $\hat{y}_0^{(t)} = \frac{y^{(t)} - \sqrt{1 - \alpha_t}\varepsilon_\theta(y^{(t)}, t)}{\sqrt{\alpha_t}}$ is the Tweedie estimate of the clean sample distribution at denoising step $t$ and $\sigma_t$ controls stochasticity [54]. Several works have leveraged the flexibility of the denoising process in performing post-hoc guidance [20] and plug-and-play generation [37, 12].

**Compositional sampling with diffusion models**. Under the diffusion model formulation, we can *compositionally* sample [11, 66] from the factor graph representation of $p(\tau)$ by calculating the score $\nabla \log p(\tau)$ as a sum of factor and variable scores following Eq. 1:

$$\nabla \log p(\tau) := \sum_{j=1}^M \nabla \log p(y_j) + \sum_{i=1}^N (1 - d_i)\nabla \log p(x_i) \tag{3}$$

In practice, our factor graph is a chain, so overlapping variables (i.e., the ones shared between neighboring factors $y_j$ and $y_{j+1}$) have degree $d_i = 2$ while non-overlapping ones have $d_i = 1$ (i.e. their marginals have no contribution to $\nabla \log p(\tau)$). For overlapping variables, we approximate the marginal scores using the average of the conditional score: $\nabla \log p(x_i) \approx \frac{1}{2}\left[\nabla \log p_{y_j}(x_i|\ldots) + \nabla \log p_{y_{j+1}}(x_i|\ldots)\right]$ where $x_i \in y_j \cap y_{j+1}$ denotes the overlapping variable between $y_j$ and $y_{j+1}$. This, along with the scores of $p(y_j)$ computed from local distribution, allows us to formulate the global compositional score $\nabla \log p(\tau)$ using Eq. 3. While this formulation enables generalization beyond the lengths seen during training, it comes with limitations described in Sec. 3.

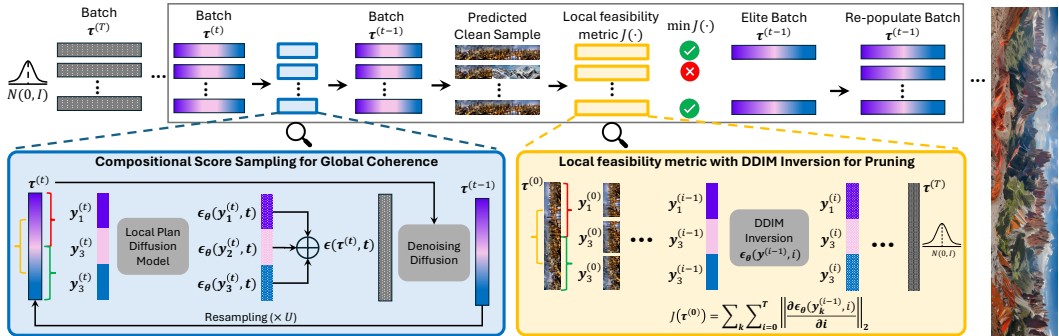

Figure 4: **Compositional diffusion with Guided Search.** At each denoising timestep, CDGS iteratively denoises a batch of noisy candidate global plans by (i) **iterative resampling** to propagate information through averaged scores at overlaps (blue) and (ii) **pruning** candidates with local inconsistencies based on the predicted clean samples (yellow). This process ensures all local plans align and belong to high-likelihood regions of $p(y)$, producing globally coherent plans.

## 3 METHOD

**Challenge: Compositional sampling with multi-modal distributions.** Solving long-horizon tasks requires constructing a coherent global plan distribution that induces an exponentially large search space and requires reasoning about long-horizon dependencies. Data scarcity prohibits directly learning the target global plan distribution $p(\tau)$, so a convincing alternative is to approximate it as a composition of local plan distribution $p(y)$ (using Eq. 1). Thus, one can sample short-horizon local plans $y_{1:M} \sim p(y)$ and compose them with suitable overlaps to form a coherent $\tau$. However, as the diversity of feasible local behaviors increases, $p(y)$ becomes highly multi-modal and composing such distributions causes $p(\tau)$ to inherit combinatorial multi-modality—where each mode of the global plan distribution corresponds to a distinct sequence of modes from the local plan distribution. In this setting, naïve compositional methods ([43]) that merge distributions $y_{1:M} \sim p(y)$ via score averaging (Eq. 3) often fail due to the mode-averaging issue: selecting high-likelihood local segments that, while individually plausible, result in incompatible mode sequences—leading to inconsistent overlaps and incoherent global plans. A natural way to address multi-modality is to explore diverse modes during sampling, an idea recently explored by inference-time scaling approaches [42, 69]. However, these methods are limited to sampling from standalone distributions and not a composed sequence of distributions. The key challenge is to generate a feasible sequence of local plans that collectively form a coherent global plan—requiring a sampling algorithm that reasons over structured combinations of modes rather than collapsing into incoherent averages.

**Our method: Compositional Diffusion with Guided Search (CDGS).** CDGS is a structured *inference-time algorithm* designed to identify coherent sequences of local modes that form valid global plans. Specifically, CDGS employs a population-based search to explore and select promising mode sequences beyond naïve sampling. To facilitate the search, it: (i) incorporates iterative resampling into the compositional score calculation to enhance information exchange across distant segments, leading to potentially coherent global plan candidates, and (ii) prunes the incoherent candidates by evaluating the likelihood of their local segments with a ranking objective. Note that this is all within a standard denoising diffusion process, making CDGS a plug-and-play sampler applicable across domains, including robotics planning, panorama image generation, and long video generation. In the following sections, we detail each of these components and demonstrate how their integration enables efficient navigation of the complex multi-modal search space to produce coherent long-horizon plans.

### 3.1 COMPOSITIONAL DIFFUSION WITH GUIDED SEARCH

A key challenge with multi-modal distributions is that naïve compositional sampling can lead to incoherent global plans: since each segment is independently sampled from $p(y)$, they may not align well at their overlaps and potentially lead to mode-averaging issues-where high-likelihood local plans do not combine to form a feasible global plan.

---

**Algorithm 1** CDGS

**Require:** Start $x_s$, Goal $x_g$, Planning horizon $H$
**Require:** Diffusion noise schedule,
**Require:** Pretrained local plan score function $\varepsilon_\theta(y^{(t)}, t)$,
**Require:** number of candidate plans $B$, number of elite plans $K$ at every step
1: Initialize $B$ global plan candidates: $\tau^{(T)}$
2: $\tau^{(T)} = (y_1^{(T)} \circ \cdots \circ y_M^{(T)}) \sim \mathcal{N}(0, \mathbf{I})$
3: **for** $t = T, \ldots, 1$ **do**
4:    $\varepsilon(\tau^{(t)}, t) = \text{ComposedScore}(\tau^{(t)}, t, \varepsilon_\theta, x_s, x_g)$
5:    $\hat{\tau}_0^{(t)} = (\tau^{(t)} - \sqrt{1-\alpha_t}\varepsilon(\tau^{(t)}, t))/\sqrt{\alpha_t}$
6:    Rank plans using $J(\hat{\tau}_0^{(t)})$ Eq. 5
7:    Select best-$K$ global plans
8:    Repopulate candidates using filtered plans
9:    $\tau^{(t-1)} \sim p(\tau^{(t-1)} | \tau^{(t)}, \hat{\tau}_0^{(t)})$ Eq. 2
10: **end for**
11: **return** $\tau^{(0)}$

---

**Algorithm 2** ComposedScore

**Require:** Noisy sample $\tau^{(t)}$, denoising timestep $t$, pretrained local plan score function $\varepsilon_\theta$
**Require:** Start and goal: $x_s, x_g$
**Require:** Number of resampling steps $U$
1: **for** $u = 1, \ldots, U$ **do**
2:    Calculate $\varepsilon(\tau^{(t)}, t)$ using Eq. 3
3:    **if** $u < U$ **then**
4:       Calculate $\tau^{(t-1)}$ using Eq. 2
5:       Add noise to $x_s/x_g$:
6:       $x_{s/g}^{(t-1)} \sim \mathcal{N}(\sqrt{\alpha_{t-1}} x_{s/g}, (1-\alpha_{t-1})I)$
7:       Inpaint noisy start and goal in $\tau^{(t-1)}$
8:       Resampling: $\tau^{(t)} \sim p(\tau^{(t)} | \tau^{(t-1)})$
9:    **end if**
10: **end for**
11: **return** $\varepsilon(\tau^{(t)}, t)$

---

To address this, our approach leverages a guided search procedure that explores promising sequences of local modes while filtering out ones that are more likely to result in incoherent global plans.

**Method formulation.** At each diffusion timestep $t$, given a noisy global plan $\tau^{(t)}$, our goal is to sample from an improved next-step distribution over $\tau^{(t-1)}$, that is more likely to yield a coherent global plan. To achieve this, we define a modified sampling distribution:

$$p_J\left(\tau^{(t-1)} | \tau^{(t)}\right) \propto p\left(\tau^{(t-1)} | \tau^{(t)}\right) \exp\left(-J\left(\hat{\tau}_0^{(t-1)}\right)/\lambda_t\right),$$

where (i) $p(\tau^{(t-1)} | \tau^{(t)})$ is the original diffusion transition realized using the compositional score function $\varepsilon(\tau^{(t)}, t)$, (ii) $\hat{\tau}_0^{(t-1)}$ is the Tweedie-estimate of the clean global plan at timestep $t-1$, (iii) $J(\cdot)$ is a plan ranking metric we define below, and (iv) $\lambda_t$ controls the exploration-exploitation tradeoff. We approximate sampling from this distribution using a Monte Carlo search procedure resembling the cross-entropy method: draw a batch of noisy global plans from $p(\tau^{(t-1)} | \tau^{(t)})$, rank them using $J$ and retain a subset of *elite* global plans that minimizes the evaluation metric $J(\cdot)$ as illustrated in Algorithm 1. The number of elites $K$ is a tunable parameter of our algorithm, enabling exploration of many possibilities in parallel when the planning problem is very large/difficult. Now, we just need to ensure that (i) the global plans are ranked appropriately and (ii) the candidate samples proposed by compositional sampling contain informative, globally coherent mode-sequences to pursue.

**Ranking global plans via local feasibility.** To guide the search effectively, we require a mechanism to evaluate the feasibility of candidate plans. Our key insight is that a global plan is feasible *iff* all of its local transitions are feasible. Since the local model $p(y)$ is trained to model feasible short-horizon behavior, high-likelihood local plans are strong indicators of local feasibility. Therefore, a globally feasible plan should consist of high-likelihood local-plan segments throughout. However, computing exact likelihoods in diffusion models is computationally expensive [55], often intractable.

To address this, we leverage DDIM inversion [54] to approximate the likelihoods of local plan segments $y$. Each local segment $y$ of a sampled global plan $\tau$ goes through forward diffusion *using the learned score network* ($\varepsilon_\theta$) such that:

$$\frac{y^{(t)}}{\sqrt{\alpha_t}} = \frac{y^{(t-1)}}{\sqrt{\alpha_{t-1}}} + \left(\sqrt{\frac{1-\alpha_t}{\alpha_t}} - \sqrt{\frac{1-\alpha_{t-1}}{\alpha_{t-1}}}\right)\varepsilon_\theta(y^{(t-1)}, t) \tag{4}$$

A high-likelihood sample follows a low-curvature path, whereas low-likelihood samples exhibit high curvature to bring noisy latents in-distribution when forward noised [18] (refer App. D). Specifically, we define a smoothness measure based on the curvature of the diffusion trajectory during inversion:

$$g\left(y^{(0)}\right) = \sum_{i=1}^{T} \left\|\frac{\partial \varepsilon_\theta(y^{(i-1)}, i)}{\partial i}\right\|_2, \quad J(\tau^{(0)}) = \prod_{m=1}^{M} \exp\left(-g\left(y_m^{(0)}\right)\right) \tag{5}$$

where $g(y^{(0)})$ measures closeness of $y^{(0)}$ to the nearest mode of $p(y)$, intuitively. A higher value of $g(y^{(0)})$ corresponds to lower-likelihood local plans. We aggregate $g(y^{(0)})$ over all local plan

segments $y_{1:M}^{(0)}$ in $\tau^{(0)}$ to define the global plan ranking metric $J(\tau^{(0)})$ to measure plan feasibility. Low-quality plans have high $J$ values, making their denoising paths more likely to be pruned.

## 3.2 ITERATIVE RESAMPLING

To ensure the effectiveness of the guided search, it is not enough to rank global plans correctly—we must also promote globally coherent candidate plans. However, standard compositional sampling fails to propagate long-horizon dependencies across overlapping local plans. Consider the running example in Fig. 3. After one denoising step, due to independent sampling of local plans, $y_1$ has no information about $y_6$, and vice versa.

To address this, we apply iterative resampling [39]: repeatedly alternating between forward noising $\tau^{(t)} \sim p(\tau^{(t)}|\tau^{(t-1)})$ and denoising steps. This procedure enables the score network's predictions for each segment to incorporate information from distant neighbors via overlapping variables, encouraging global consistency. Mathematically, this process resembles belief propagation on a chain of factors where each local plan $y_m \in y_{1:M}$ in $\tau$ depends on its neighbors $y_{m-1}$ and $y_{m+1}$ through the respective overlaps ($y_m \cap y_{m-1}$ and $y_m \cap y_{m+1}$). During resampling, the belief of $y_m$ is updated as: $p(y_m|y_{m-1}, y_{m+1}) \propto p(y_m)p(y_m|y_m \cap y_{m-1})p(y_m|y_m \cap y_{m+1})$ Following Algorithm 2, after $U$ iterations, this iterative resampling ensures that information propagates across the entire long-horizon sequence, producing a more globally coherent plan.

**Summary of CDGS.** We propose a guided-search algorithm by integrating a population-based pruning strategy within compositional sampling. Given a local plan score function, our approach samples potentially coherent global plan candidates and filters out plans with locally inconsistent segments. Repeating this throughout the denoising process improves the probability that the retained candidates satisfy local feasibility at every segment and are therefore globally feasible plans. Our algorithm benefits from adaptive compute at inference time, with the flexibility to scale the batch size $B$ and the number of resampling steps $U$ for problems with longer horizons and larger search spaces.

## 4 EXPERIMENTAL RESULTS: ROBOTIC PLANNING

In this section, we evaluate the performance of CDGS for long-horizon robotic planning. For all the experiments, we represent inputs with a low-dimensional state-space of the system comprising the pose of the end-effector and the objects in the scene in the global frame of reference. For real-world evaluations, we obtain the pose of the objects through perception, more details in App. H.

**CDGS can solve learning from play and stitching problems efficiently.** We evaluate CDGS for sequential-decision making tasks using the OGBench Maze and Scene task suite [48], which includes PointMaze and AntMaze along with five tasks for Scene where a robot must manipulate objects (a drawer, sliding window, and cube) to reach a goal state. The primary challenge is learning from *small maze trajcetories* or *unstructured play data* during training, which does not directly solve the target tasks. The diversity of the unstructured plans makes the local distributions highly multimodal. We hypothesize that CDGS is an ideal method for this problem statement because it can compose short-horizon plans into meaningful long-horizon plans.

CDGS uses a Diffuser [24] to learn the distribution of local plans (up to 4 secs of trajectory at 20 Hz) represented as a sequence of states and actions $y = \{s_1, a_1, ..., s_h, a_h\}$ and then composes them at inference for a given goal state to sample up to 10 secs of motion plans $\tau = \{s_i, a_i\}_{i=1}^{H}$ ($h < H$). We compare the performance of CDGS with inverse reinforcement learning baselines from OGBench, including GCBC [41, 17], GCIVL, GCIQL [31], and HIQL [49], with results presented in Tab. 1. In addition we also include compositional generative baselines like GSC [43] and CompDiffuser [40]. It should be noted that CDGS with resampling and pruning can scale the performance of naïve compositional sampling (GSC), in a training-free manner, to an extent that beats baselines like CompDiffuser [40] that use overlap information while training and learn an overlap conditioned score function. Finally, we also validate CDGS on composite ball reaching and ball carrying trajectory stitching of AntSoccer in OGBench and show the results in Tab. 2.

**CDGS can solve hybrid-planning problems.** Task and Motion Planning (TAMP) decomposes robotic planning into a symbolic search for a sequence of discrete high-level skills (e.g., `pick`, `place`, `pull`) followed by low-level motion planning for each skill [14]. Specifically, we formu-

| Environment | Type | Size | GCBC | GCIVL | GCIQL | QRL | CRL | HIQL | GSC | CD | Ours w/o PR | Ours |
|---|---|---|---|---|---|---|---|---|---|---|---|
| **PointMaze** | Stitch | Medium | 23 ±18 | 70 ±14 | 21 ±9 | 0 ±1 | 80 ±12 | 74 ±6 | **100** ±0 | 100 ±0 | 100 ±0 | **100** ±0 |
| | | Large | 7 ±5 | 12 ±6 | 31 ±2 | 0 ±0 | 84 ±15 | 13 ±6 | 100 ±0 | 100 ±0 | 100 ±0 | **100** ±0 |
| | | Giant | 0 ±0 | 0 ±0 | 0 ±0 | 0 ±0 | 50 ±8 | 0 ±0 | 29 ±2 | 68 ±3 | 78 ±2 | **87** ±3 |
| **AntMaze** | Stitch | Medium | 45 ±11 | 44 ±6 | 29 ±6 | 59 ±7 | 53 ±6 | 94 ±1 | **97** ±2 | 96 ±2 | 97 ±2 | **97** ±1 |
| | | Large | 3 ±3 | 18 ±2 | 7 ±2 | 18 ±2 | 11 ±2 | 67 ±5 | 66 ±2 | 86 ±2 | 86 ±2 | **88** ±2 |
| | | Giant | 0 ±0 | 0 ±0 | 0 ±0 | 0 ±0 | 0 ±0 | 21 ±2 | 20 ±1 | 65 ±3 | 82 ±1 | **85** ±3 |
| **Humanoid-Maze** | Stitch | Medium | 29 ±5 | 12 ±2 | 12 ±3 | 18 ±2 | 71 ±3 | **96** ±4 | 92 ±1 | 91 ±1 | 88 ±3 | 93 ±1 |
| | | Large | 6 ±3 | 1 ±1 | 0 ±0 | 3 ±1 | 6 ±1 | 31 ±3 | 70 ±3 | 72 ±3 | 70 ±2 | **74** ±2 |
| | | Giant | 0 ±0 | 0 ±0 | 0 ±0 | 0 ±0 | 0 ±0 | 12 ±2 | 5 ±1 | **64** ±4 | 47 ±5 | 55 ±3 |
| **Scene** | Play | - | 5 ±1 | 42 ±4 | **51** ±4 | 5 ±1 | 19 ±2 | 38 ±3 | 8 ±2 | 13 ±1 | 36 ±6 | **51** ±2 |

Table 1: **OGbench [48]: learning from stitch and play datasets.** With much less training data requirements, CDGS performs on-par with inverse-reinforcement learning baselines and better than generative baselines in a receding horizon control. For GSC, CD and CDGS, we replan based on distance from goal for maze tasks (following CD [40]) and sample the complete plan based on the oracle planning horizon for scene task. Success rate averaged over 100 trials and 3 seeds with randomly chosen task ids. Baseline performance is borrowed from original papers [48, 40]

| Environment | Size | GCBC | GCIVL | GCIQL | QRL | CRL | HIQL | GSC (4D) | CD (4D) | GSC (17D) | CD (17D) | Ours w/o PR (17D) | Ours (17D) |
|---|---|---|---|---|---|---|---|---|---|---|---|---|---|
| **AntSoccer** | Arena | 34 ±4 | 21 ±3 | 5 ±2 | 2 ±1 | 2 ±1 | 23 ±2 | 41 ±4 | 55 ±6 | 65 ±3 | **69** ±3 | 67 ±3 | **69** ±1 |
| | Medium | 2 ±1 | 1 ±0 | 0 ±0 | 0 ±0 | 0 ±0 | 8 ±2 | 5 ±2 | 13 ±1 | 12 ±2 | 17 ±3 | 16 ±3 | **18** ±2 |

Table 2: **Stitching composite task AntSoccer in OGBench [48].** We evaluate CDGS on high-dimensional (17D) state space to stitch ball reaching and ball carrying behaviors for two AntSoccer environments. Success rate averaged over 100 trials and 3 seeds with randomly chosen task ids. Baseline performance is borrowed from original papers [48, 40]

late a task-and-motion-plan as $\tau = \{y_1, \ldots, y_m\}$ where $y_i = \{s_i, \pi_i, a_i, s_{i+1}\}$ where a discrete skill $\pi_i$ with motion parameters $a_i$ is executed on state $s_i$ to get to state $s_{i+1}$. This entails solving a hybrid-planning problem where the chosen discrete skill modes and continuous action modes must simultaneously satisfy symbolic and geometric constraints. We systematically evaluate our method on three suites of TAMP tasks, which are described in detail in App. E.

We compare our method with learning-based TAMP and other compositional methods. Specifically, we consider the following categories: (1) privileged with Planning Domain Definition Language (PDDL): **Random CEM** and **STAP CEM** [2] search for symbolic plans in a manually and systematically constructed PDDL domain and apply Cross Entropy Method (CEM) optimization over potential motion plans (2) task information provided via prompting: **LLM-T2M** [36] prompts an LLM (GPT-4.1) and VLM (**VLM-T2M**) with descriptions of the scene along with $n = 11$ in-context examples (w/o and w/ scene images respectively) to generate a feasible task plan that is checked by a geometric motion planner (STAP CEM in this case). (3) compositional diffusion: **GSC (no task plan)** [43] performs compositional diffusion (equivalent to CDGS w/o RP and PR). Notably, GSC and CDGS are the *only* methods that do not rely on explicit symbolic search or

| | Remark (Task information) | Hook Reach | | Rearrangement Push | | Rearrangement Memory | |
|---|---|---|---|---|---|---|---|
| | | Task 1 | Task 2 | Task 1 | Task 2 | Task 1 | Task 2 |
| Task Length | | 4 | 5 | 4 | 7 | 4 | 7 |
| Random CEM | PDDL + BFS | 0.14 | 0.10 | 0.08 | 0.00 | 0.02 | 0.02 |
| STAP CEM | | **0.66** | **0.70** | 0.76 | **0.70** | 0.00 | 0.00 |
| LLM-T2M, $n = 11$ | LM Prior + | 0.0 | 0.48 | 0.72 | 0.06 | 0.0 | 0.0 |
| VLM-T2M, $n = 11$ | Prompting | 0.0 | 0.42 | 0.62 | 0.02 | 0.0 | 0.0 |
| GSC (Original) | Oracle task plan | 0.78 | 0.80 | 0.88 | 0.64 | 0.82 | 0.48 |
| GSC (no task plan) | No PDDL skill-level data only | 0.18 | 0.04 | 0.00 | 0.00 | 0.07 | 0.00 |
| CDGS (w/o PR) | | 0.24 | 0.12 | 0.12 | 0.00 | 0.11 | 0.00 |
| CDGS (ours) | | **0.64** | 0.58 | **0.84** | 0.48 | **0.42** | **0.18** |

Table 3: **Evaluation on TAMP task-suite.** We compare CDGS with relevant search-based (PDDL Domain) and prompting based (LLM/VLM) baselines. CDGS performs on-par or slightly trails privileged methods on Hook Reach and Rearrangement Push, but substantially outperforms them on Rearrangement Memory. (success rate over 50 trials)

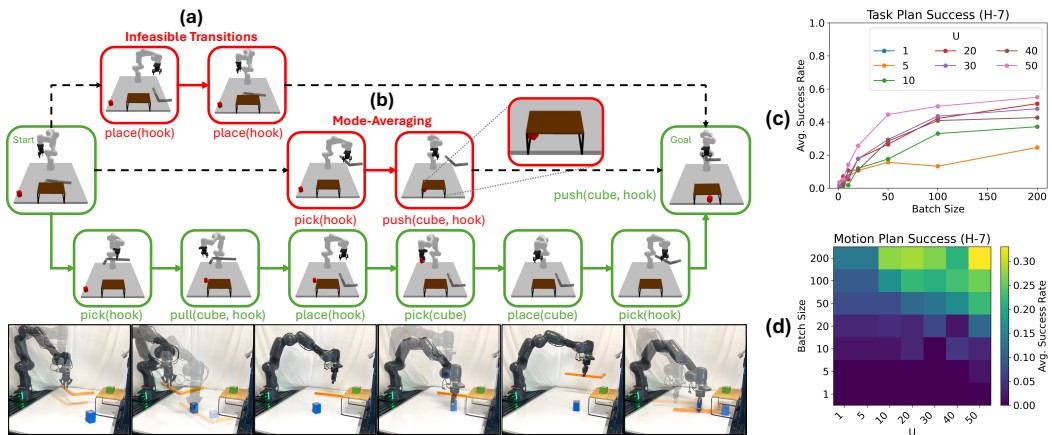

Figure 5: **Left: Visualizing plan pruning.** When compositional sampling chooses an infeasible mode sequence, the resulting plan can hallucinate out-of-distribution transitions due to **mode-averaging** as explained in Sec. 3. For instance, **(a) Infeasible transitions:** `inhand(hook)` precondition is never met for `place(hook)`, and **(b) State hallucination:** `cube` moves `under(rack)` as a result of averaging toward the goal state, despite being geometrically infeasible for `push(cube, hook)`. Our pruning objective (Eq. 5) ensures only feasible plans during denoising, where all transitions are in-distribution with our short-horizon transition diffusion model. **Right: Scaling analysis.** (H-7) denotes performance averaged over tasks of horizon 7. **(c) Task planning** success improves with batch size, with larger gains from more resampling steps. **(d) Motion planning** success improves with resampling steps, but only when batch size is large enough

LLM/VLM supervision for the task plan. The results of our evaluation are in Tab. 3. Note that while **GSC (Original)** [43] leverages skill-level expert diffusion models and oracle task plan, in our case it represents naïve compositional sampling with a unified model (w/o oracle task plan).

**CDGS's performance scales with compute.** We hypothesize that CDGS has adaptive inference-time compute, meaning that it benefits from more compute on harder problems. We validate this hypothesis on our most challenging TAMP tasks with a planning horizon of 7. We find that increasing batch size ($B$) and number of resampling steps ($U$) increases the task planning success Fig. 5(c) and motion planning success Fig. 5(d) of CDGS. Interestingly, we find that neither increasing $B$ nor $U$ on their own is sufficient for overall motion planning success. Thus, both resampling and pruning are essential for long-horizon tasks, as evidenced by the significant improvement of CDGS (Tab. 3).

## 5 CDGS FOR LONG CONTENT GENERATION

We formulate CDGS with specific design choices that enable (i) efficient message passing for global consistency and (ii) pruning denoised paths that lead to incoherent sequences. While these mechanisms are essential for long-horizon planning, we investigate their broader applicability, particularly in long-content generation tasks such as text-to-image (T2I) and text-to-video (T2V), which require spatial and temporal coherence over extended horizons. Our framework demonstrates effective improvement in long-horizon content generation.

**CDGS enables coherent panoramic image generation via stitching.** We evaluate CDGS on panoramic synthesis by composing multiple image patches. A panorama $\tau$ is represented as a sequence of small images $y$, each split into three overlapping patches $y = (x_1, x_2, x_3)$. Using Stable Diffusion-2.0 [51], we generate up to $512 \times 4608$ panoramas by stitching $512 \times 512$ images. We compare against (i) Multi-Diffusion (MD) [4], which averages scores across overlaps (image-domain analogue of GSC [43]), and (ii) Sync-Diffusion (SD) [33], which enforces LPIPS-based perceptual guidance [67]. As shown in Tab. 4, CDGS matches SD without explicit perceptual loss, indicating effective message passing for global style and perceptual transfer while maintaining prompt alignment (CLIP [50]). Qualitative samples are shown in Fig. 6, with more details in App. A.

**CDGS can sample temporally-consistent longer videos.** We follow a setup similar to panorama generation, composing shorter clips along the temporal axis for long-video generation. When short

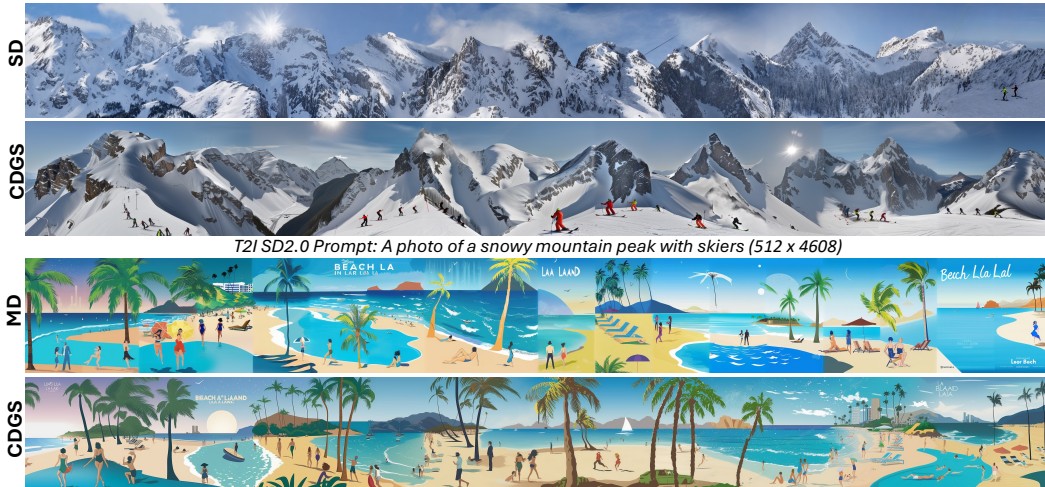

*T2I SD2.0 Prompt: A photo of a snowy mountain peak with skiers (512 x 4608)*

*T2I SD2.0 Prompt: An illustration of a beach in La La Land style (512 x 4608)*

Figure 6: **Panorama image generation.** The above figure shows the qualitative comparison of CDGS with MD [4] and SD [33]. We show qualitative intuition behind global coherence and local feasibility: while SD generates smooth panoramas, they fail to satisfy the global context (*mountain peak with skiers*), on the other hand, MD follows the global context (*beach in La La Land style*) but fails to exhibit local consistency. CDGS excels at both.

| Metric | GSC/Multi-Diffusion | Sync-Diffusion | CDGS w/o PR | CDGS |
|---|---|---|---|---|
| Intra-LPIPS $\downarrow$ | 0.72 $_{\pm 0.08}$ | **0.58** $_{\pm 0.06}$ | 0.61 $_{\pm 0.08}$ | **0.59** $_{\pm 0.04}$ |
| Intra-Style-L$_{(\times 10^{-2})}$ $\downarrow$ | 2.96 $_{\pm 0.24}$ | **1.39** $_{\pm 0.12}$ | 1.97 $_{\pm 0.08}$ | **1.38** $_{\pm 0.03}$ |
| Mean-CLIP-S $\uparrow$ | 31.77 $_{\pm 2.14}$ | 31.77 $_{\pm 2.14}$ | 31.71 $_{\pm 2.34}$ | **32.51** $_{\pm 2.66}$ |

Table 4: **Quantitative comparison of panorama generation.** We generate 1000 panoramas of dimensions $512 \times 4608$ using 14 prompts and compare different methods based on their perceptual similarity (LPIPS [67]), style similarity (Style-loss [15]), and prompt alignment (CLIP score [50]).

sequences of frames are stitched to make a long video, a key challenge is maintaining subject consistency and minimizing temporal artifacts. We use CogVideoX-2B [62] as the base model, capable of generating $\sim 50$-frame videos, and extend it to up to 350 frames at 720p resolution. We use six prompts to generate videos with naïve composition (GSC/Gen-L-Video [56] equivalent), compositional diffusion with resampling, and CDGS. The results are evaluated with VBench [22] for temporal consistency, subject fidelity, visual quality, and alignment with the prompt (refer Tab. 5). Qualitative analysis in Fig. 7 clearly shows the multimodal problem where multiple local plans allow satisfying the global context, but with CDGS's effect local-to-global message passing, we see an improvement in subject consistency and temporal smoothness. This comes at a minor aesthetic degradation—a tradeoff commonly observed in long-video generation models.

| Method | Subject-consistency $\uparrow$ | Temporal-flickering $\uparrow$ | Aesthetic-quality $\uparrow$ | Prompt-alignment $\uparrow$ |
|---|---|---|---|---|
| CogVideoX-2B (50 frames) | **95.91** | **97.35** | **63.10** | **25.51** |
| CogVideoX-2B (350 frames) | 90.24 | 98.44 | 49.44 | 21.78 |
| GSC ($\equiv$ Gen-L-Video) | 89.51 | 96.89 | **60.12** | 25.13 |
| Ours w/o PR | 91.06 | 97.08 | 59.40 | 25.42 |
| Ours | **91.67** | 97.16 | 58.90 | **26.13** |

Table 5: **Quantitative comparison of long-video generation.** We evaluate the performance of CDGS based on selected metrics from VBench that measure subject consistency, aesthetics, prompt alignment and temporal artifacts. We use 6 prompts (refer App. B) and generate videos with 350 frames at 720p resolution. CDGS achieves competitive video quality but for significantly (7x) extended horizons.

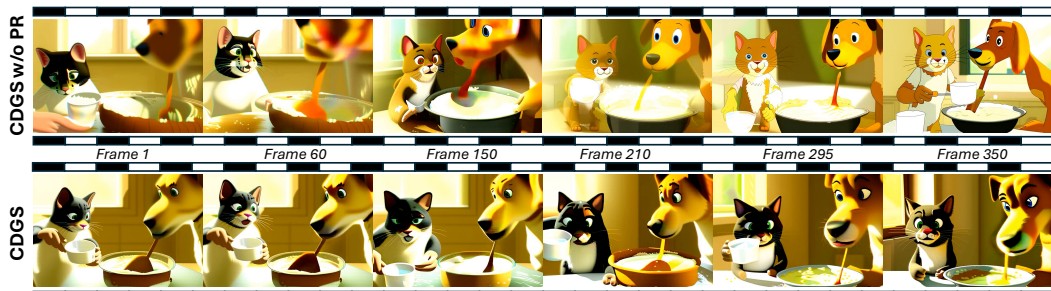

*T2V CogVideoX-2b Prompt: A cat and a dog baking a cake together in a kitchen. The cat is carefully measuring flour, while the dog is stirring the batter with a wooden spoon. The kitchen is cozy, with sunlight streaming through the window. (350 frames, 720p)*

Figure 7: **Long video generation.** CDGS w/ PR (below) maintains subject-consistency while CDGS w/o PR (top) exhibits mode-averaging, resulting in significant changes to the subjects' appearances.

## 6 RELATED WORK

**Long-horizon content generation** There are many approaches to generating long-horizon content like panoramas and long videos [26, 38, 7, 19]. Some assume access to long-horizon training data for end-to-end training [16, 5, 60, 61], while others with weaker assumptions about training data will compose the outputs of short-horizon models through outpainting [59, 28] or stitching [66, 29, 34, 32, 47, 6, 40]. Our method belongs to the latter, enabling generalization to longer horizons than seen during training.

**Generative planning.** Generative models such as diffusion models[53, 21] are widely used for planning [25, 3, 8, 35, 40], though they struggle with task lengths beyond their training data. Recent works including Diffusion-CCSP [63], GSC [44], and GFC [45] have explored compositional sampling [37, 12, 66] but they sidestep the mode-averaging problem via additional mode supervision in the form of task skeletons or constraint graphs. In contrast, our approach directly addresses the mode-averaging problem to generate goal-directed long-horizon plans from short-horizon models.

**Inference-time compute.** Scaling inference-time computation is a powerful strategy for improving the performance of generative models [58, 46]. For diffusion models [54, 27], recent work has shown the efficacy of scaling inference-time compute through verifier-guided search during the denoising process [42, 52, 65, 68, 69]. Our algorithm differs in that it addresses the unique limitation of mode-averaging when sampling from a compositional chain of distributions.

## 7 CONCLUSION

We introduce CDGS, a framework integrating compositional diffusion with guided search to generate long-horizon sequences with short-horizon models. By embedding search within the denoising process, CDGS can handle composing highly multimodal distributions and sample solutions that are both globally coherent and locally feasible. Qualitative and quantitative results suggest that CDGS is a general pathway for extending the reach of generative models beyond their training horizons across robotic planning, panoramic images, and video generation.

## 8 LIMITATIONS

While CDGS demonstrates strong performance in long-horizon goal-directed planning, it relies on a few simplifying assumptions that also suggest directions for future work. We assume the ability to specify a goal state, which simplifies planning but can be naturally extended to goal-generation or classifier-guided goal-conditioning methods [13]. Similarly, we generate plans for a fixed horizon, yet the framework can handle arbitrary horizons given the same start and goal, enabling selection among multiple candidate plan lengths. Finally, long-horizon dependencies are communicated through score averaging and resampling between adjacent skills; more sophisticated message-passing or attention-based mechanisms could improve efficiency and coherence across entire plans. These assumptions keep the problem tractable while providing a flexible foundation for extending CDGS to more general and complex planning scenarios.

## 9 REPRODUCIBILITY STATEMENT

We are committed to ensuring that all the results presented in this paper are reproducible. To this end, we have provided pseudocodes in the paper and released the official code base through our project website: `https://cdgsearch.github.io/`. We have also provided the hyperparameters table for motion planning (refer App. F), for image generation (refer App. J) and video generation (refer App. K). Apart from this our content-generation experiments use open-source models like Stable-Diffusion-2 (refer `https://huggingface.co/stabilityai/stable-diffusion-2`) and CogVideoX-2B (refer `https://huggingface.co/zai-org/CogVideoX-2b`). For all other robotics setup, we provide more information through appendix and our project website.

## 10 LLM USAGE

LLMs were not used in any manner for conceptualization of the idea, key contributions of the proposed work and finding relevant prior woks.

## ACKNOWLEDGMENTS

This work is partially supported by NSF-2442393, NSF-2409016, NSF-1942523, and Samsung Research America.

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
