## CONTENTS

# A ADDITIONAL PANORAMA GENERATION RESULTS

*A photo of mountain range at twilight*

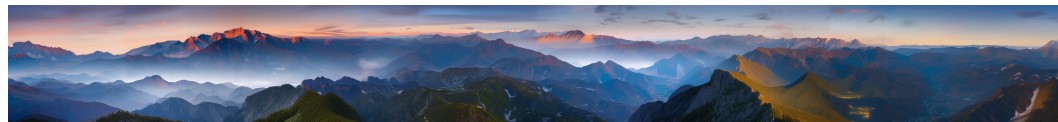

*A photo of a grassland with animals*

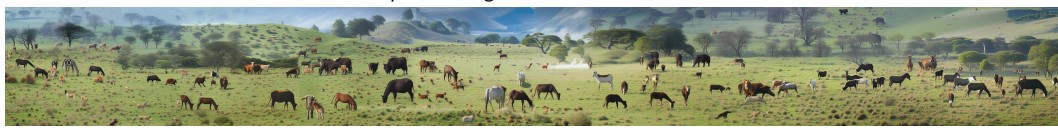

*Silhouette wallpaper of a dreamy scene with shooting stars*

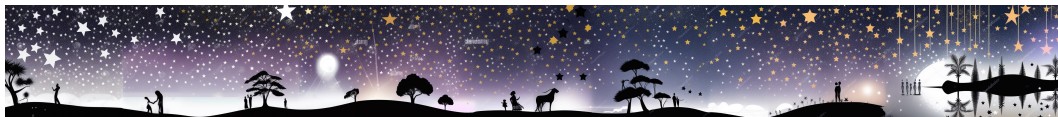

*Natural landscape in anime style illustration*

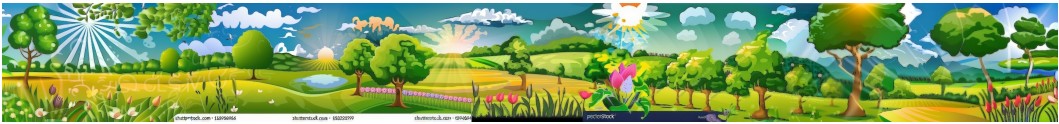

*A photo of a beautiful ocean with coral reef*

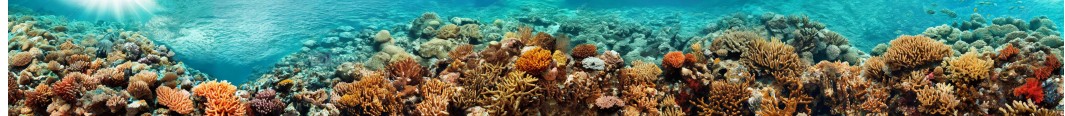

*A photo of a lake under the northern lights*

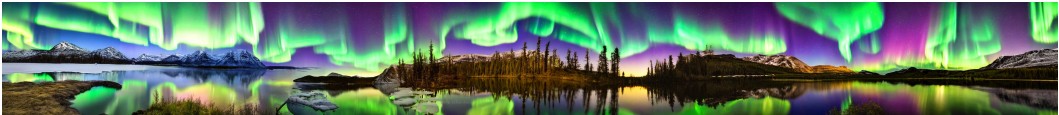

*A beautiful landscape with mountains and a river*

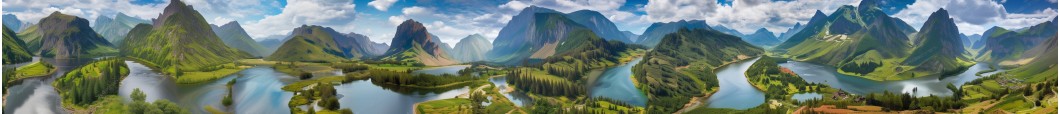

*Last supper with cute corgis*

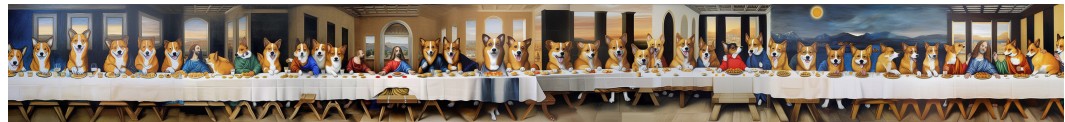

*A photo of a forest with a misty fog*

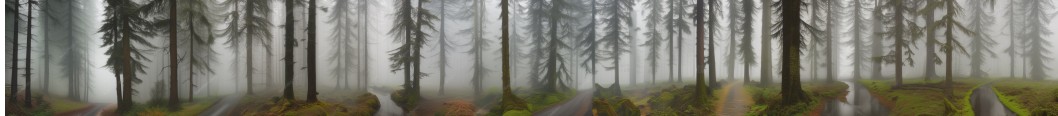

*A photo of a rock concert*

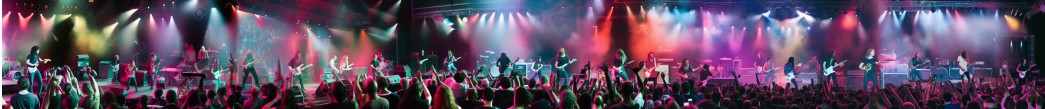

## B    PROMPTS FOR VIDEO GENERATION

We used the following standard prompts for generating the videos:

1. The camera follows behind a white vintage SUV with a black roof rack as it speeds up a steep dirt road surrounded by pine trees on a steep mountain slope, dust kicks up from it's tires, the sunlight shines on the SUV as it speeds along the dirt road, casting a warm glow over the scene. The dirt road curves gently into the distance, with no other cars or vehicles in sight. The trees on either side of the road are redwoods, with patches of greenery scattered throughout. The car is seen from the rear following the curve with ease, making it seem as if it is on a rugged drive through the rugged terrain. The dirt road itself is surrounded by steep hills and mountains, with a clear blue sky above with wispy clouds. realism, lifelike.

2. A cute happy panda, dressed in a small, red jacket and a tiny hat, sits on a wooden stool in a serene bamboo forest. The panda's fluffy paws strum a miniature acoustic guitar, producing soft, melodic tunes, move hands, singings. Nearby, a few other pandas gather, watching curiously and some clapping in rhythm. Sunlight filters through the tall bamboo, casting a gentle glow on the scene. The panda's face is expressive, showing concentration and joy as it plays. The background includes a small, flowing stream and vibrant green foliage, enhancing the peaceful and magical atmosphere of this unique musical performance. realism, lifelike.

3. A group of colorful hot air balloons take off at dawn in Cappadocia, Turkey. Dozens of balloons in various bright colors and patterns slowly rise into the pink and orange sky. Below them, the unique landscape of Cappadocia unfolds, with its distinctive 'fairy chimneys' - tall, cone-shaped rock formations scattered across the valley. The rising sun casts long shadows across the terrain, highlighting the otherworldly topography. realism, lifelike.

4. A detailed wooden toy ship with intricately carved masts and sails is seen gliding smoothly over a plush, blue carpet that mimics the waves of the sea. The ship's hull is painted a rich brown, with tiny windows. The carpet, soft and textured, provides a perfect backdrop, resembling an oceanic expanse. Surrounding the ship are various other toys and children's items, hinting at a playful environment. The scene captures the innocence and imagination of childhood, with the toy ship's journey symbolizing endless adventures in a whimsical, indoor setting. realism, lifelike.

5. A young woman with beautiful and clear eyes and blonde hair standing and white dress in a forest wearing a crown. She seems to be lost in thought, and the camera focuses on her face. The video is of high quality, and the view is very clear. High quality, masterpiece, best quality, highres, ultra-detailed, fantastic. realism, lifelike.

6. A woman walks away from a white Jeep parked on a city street at night, then ascends a staircase and knocks on a door. The woman, wearing a dark jacket and jeans, walks away from the Jeep parked on the left side of the street, her back to the camera; she walks at a steady pace, her arms swinging slightly by her sides; the street is dimly lit, with streetlights casting pools of light on the wet pavement; a man in a dark jacket and jeans walks past the Jeep in the opposite direction; the camera follows the woman from behind as she walks up a set of stairs towards a building with a green door; she reaches the top of the stairs and turns left, continuing to walk towards the building; she reaches the door and knocks on it with her right hand; the camera remains stationary, focused on the doorway; the scene is captured in real-life footage.

7. At sunset, a modified Ford F-150 Raptor roared past on the off-road track. The raised suspension allowed the huge explosion-proof tires to flip freely on the mud, and the mud splashed on the roll cage.

8. A cat and a dog baking a cake together in a kitchen. The cat is carefully measuring flour, while the dog is stirring the batter with a wooden spoon. The kitchen is cozy, with sunlight streaming through the window.

## C COMPOSITIONAL SCORE COMPUTATION: CDGS'S RELATION TO EXISTING LITERATURE

Composing the distributions defined by multiple diffusion models is well-explored in literature [12, 63]. Specifically we want to sample from the distribution of long-horizon sequences $\tau = (x_1, x_2, ..., x_N)$ by composing distributions of short-horizon sequences. There have been two main ways of composing short-horizon diffusion models in a chain:

1. Score-Averaging: approaches like GSC [44] and CDGS partition $\tau$ into overlapping segments where the score for regions of overlap can be obtained by score-averaging:

$$p(\tau) \propto \frac{p(x_1, x_2, x_3) p(x_3, x_4, x_5) \dots}{p(x_3) \dots}$$

2. Conditioning: CompDiffuser [40] partitions $\tau$ into non-overlapping segments that are conditioned on adjacent segments

$$p(\tau) \propto p(x_1|x_2) p(x_N|x_{N-1}) \prod_{i=2}^{N-1} p(x_i|x_{i-1}, x_{i+1})$$

Since CompDiffuser [40] requires training a model with conditions, we follow the more plug-n-play format of GSC [43]. For TAMP, the key difference between CDGS and GSC is that individual skill-level transitions for GSC are already conditioned on the task plan. This means that CDGS samples from the unified model $p(s_{i-1}, a_i, s_i)$ where for GSC individual segments are sampled from $p(s_{i-1}, a_i, s_i | \pi_i)$ since the oracle skill-sequence (task plan) $\pi_{1:H}$ is already provided. This greatly simplifies compositional sampling as the models in GSC only conduct motion planning, thus reducing multi-modality and mode-averaging issues significantly, whereas the models in CDGS conduct full task and motion planning.

## D PRUNING OBJECTIVE VIA DDIM INVERSION: CDGS'S RELATION TO EXISTING LITERATURE

DDIM Inversion is simply running the DDIM [54] denoising process backward i.e., forward noising in a deterministic way, to extract the denoising path from clean samples. Since we sample plans from a composed distribution, transition segments of a good plan should follow high-likelihood regions of the unified skill-transition distribution. A DDIM sampling based denoising looks like:

$$x^{(t-1)} = \sqrt{\alpha_{t-1}} \left( \frac{x^{(t)} - \sqrt{1 - \alpha_t} \varepsilon_\theta(x_t, t)}{\sqrt{\alpha_t}} \right) + \sqrt{1 - \alpha_{t-1}} \varepsilon_\theta(x^{(t)}, t)$$

We follow [18] to formulate this metric by first forward-noising each segment of the sampled plan from the task-level distribution according to:

$$\frac{x^{(t)}}{\sqrt{\alpha_t}} = \frac{x^{(t-1)}}{\sqrt{\alpha_{t-1}}} + \left( \sqrt{\frac{1 - \alpha_t}{\alpha_t}} - \sqrt{\frac{1 - \alpha_{t-1}}{\alpha_{t-1}}} \right) \varepsilon_\theta(x^{(t-1)}, t)$$

With $\delta_t = \sqrt{\frac{1 - \alpha_t}{\alpha_t}}$ and $y^{(t)} = x^{(t)} \sqrt{1 + \delta_t^2}$, we can convert the above into:

$$dy_t = \varepsilon_\theta(x^{(t-1)}, t) d\delta_t$$

Lets consider two forward-noising paths from two samples: one from high-likelihood region and one from a low-likelihood region. For both the samples, the rate of change of the integration path and its curvature directly indicate the likelihood of the clean sample. A high-likelihood sample will follow a smoother path with less curvatures while a low -likelihood sample will follow a high-curvature path to bring the noisy samples to high-likelihood regions of the noisy distribution. Hence, we consider

Taylor expansion to analyze the higher order terms:

$$y^{(t+1)} = y^{(t)} + (\delta_{t+1} - \delta_t)\frac{dy^{(t)}}{d\delta_t}\big|_{(y^{(t)},t)} + (\delta_{t+1} - \delta_t)^2 \frac{d^2 y^{(t)}}{d\delta_t^2}\big|_{(y^{(t)},t)} + \ldots$$

$$= y^{(t)} + (\delta_{t+1} - \delta_t)\varepsilon_\theta(x^{(t-1)},t) + (\delta_{t+1} - \delta_t)^2 \frac{d\varepsilon_\theta(x^{(t-1)},t)}{d\delta_t}\big|_{(y^{(t)},t)} + \ldots$$

where the second derivative term can be further decomposed into

$$\frac{d\varepsilon_\theta(x^{(t-1)},t)}{d\delta_t} = \frac{\partial\varepsilon_\theta(x^{(t-1)},t)}{\partial x^{(t-1)}}\frac{dx^{(t-1)}}{d\delta_t} + \frac{\partial\varepsilon_\theta(x^{(t-1)},t)}{\partial t}\frac{dt}{d\delta_t}$$

We find that the time-derivative term $\dfrac{\partial\varepsilon_\theta(x^{(t-1)},t)}{\partial t}$ is sufficient to distinguish between denoising path from high and low likelihood samples. Thus, we construct our pruning objective as:

$$g(x^{(0)}) = \sum_{t=1}^{T} \left\| \frac{\partial\varepsilon_\theta(x^{(t-1)},t)}{\partial t} \right\|_2$$

which is summing the curvature of the complete denoising timestep. A lower value of $g(x_0)$ indicates high-likelihood samples. The final objective of a sampled plan $\tau$ composing of segments $(x_1, x_2, \ldots, x_H)$, where $x_k = (s_{k-1}, \pi_{k-1}, a_{k-1}, s_k)$, is calculated as:

$$\prod_{k=1}^{H} \exp\left(-g(x_k^{(0)})\right)$$

Based on the cumulative score of all segments of a plan, we select top-M plans to move on to the next denoising timestep of the compositional sampling process.

In this section, we want to understand the efficacy of the DDIM inversion based pruning objective.

### D.1 ILLUSTRATIVE EXAMPLE: DDIM INVERSION AND OOD METRICS

**Experiment description:** We learn a 1D distribution of $x$ such that $[-1.0, -0.5] \cup [-0.1, 0.2] \cup [0.6, 1.0]$ is in-distribution (ID) and remaining segments are out of distribution (OOD) by construction. We learn a simple MLP score function to represent the diffusion model.

We draw clean samples uniformly from $[-1.0, 1.0]$ and use DDIM inversion with the learned score function to noise them for 100 timesteps and then use 100 steps of DDIM denoising to reconstruct the clean samples back. Note that the original clean samples contain both ID and OOD while the reconstructed samples only contain ID.

We calculate the following metrics:

1. DDIM inversion metric: This is what is used in CDGS. The goal is to quantify the curvature of the inversion path. Smoother path means high-likelihood clean sample, while a path with abrupt direction changes mean low-likelihood clean samples. We only measure the cummulative curvature of the first 20% inversion trajectory as, after that the path stabilizes as noisy latents come within in-distribution regions.

2. Reconstruction metric: We calculate the error between the reconstructed sample and the clean sample. Note that this is after 100 steps of inversion followed by 100 denoising steps as shown in Fig. 8.

3. Restoration Gap: This is another form of reconstruction metric but we do not need to inversion to obtain the noisy latents. We can sample any denoising timestep, add noise to the timestep using $x_t = \sqrt{\alpha_t}x_0 + \sqrt{1-\alpha_t}\varepsilon$, $\quad \varepsilon \in N(0,I)$ and then denoise $x_t$ from timestep $t$ to obtain reconstructed clean sample $\hat{x}_0^t$. Thus restoration gap can be calculated as: $\mathbb{E}_t[\hat{x}_0^t - x_0]$. This can be repeated for multiple choice of timesteps.

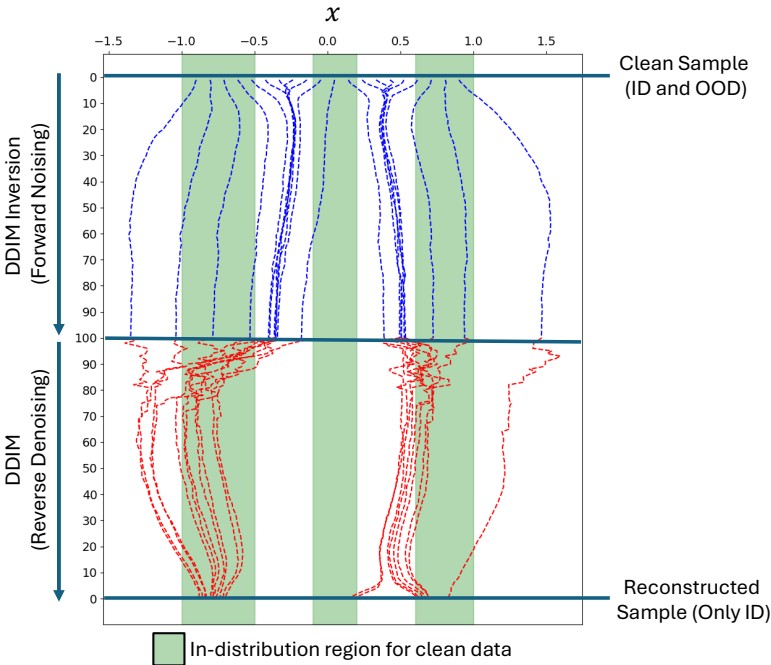

Figure 8: This plot contrasts the DDIM Inversion (forward noising, blue lines) with DDIM Denoising (reverse, red lines) on a 1D dataset where green areas mark the in-distribution (ID) regions. **Top.** (Inversion, $t = 0 \rightarrow 100$) shows both ID and out-of-distribution (OOD) clean samples diffusing into noise using the learned score function. **Bottom.**(Denoising, $t = 100 \rightarrow 0$) illustrates the learned model starting from noise and guiding all trajectories to reconstruct samples only within the valid ID regions, demonstrating how OOD paths are pulled back to the data manifold.

**Advantages of the curvature-based approach over reconstruction-based alternatives for likelihood approximation:** We see two directions of improvement when using CDGS's curvature-based metric vs reconstruction-based alternatives:

1. DDIM inversion only requires forward noising while reconstruction methods require both forward noising and denoising back.

2. For distributions with disjoint modes (like the one considered for this experiment), it is not necessary that the reconstructed sample after noising and denoising will belong to the same mode as the original clean sample. This makes reconstruction-based metrics invalid or overly conservative, neglecting in-distribution segments. We show this in Fig. 9 where the ID samples from middle segment after reconstruction belong to the left and right segments. While this increases the reconstruction error, the curvature metric can robustly handle this phenomenon. On the other hand, the restoration gap fails to give any meaningful signal.

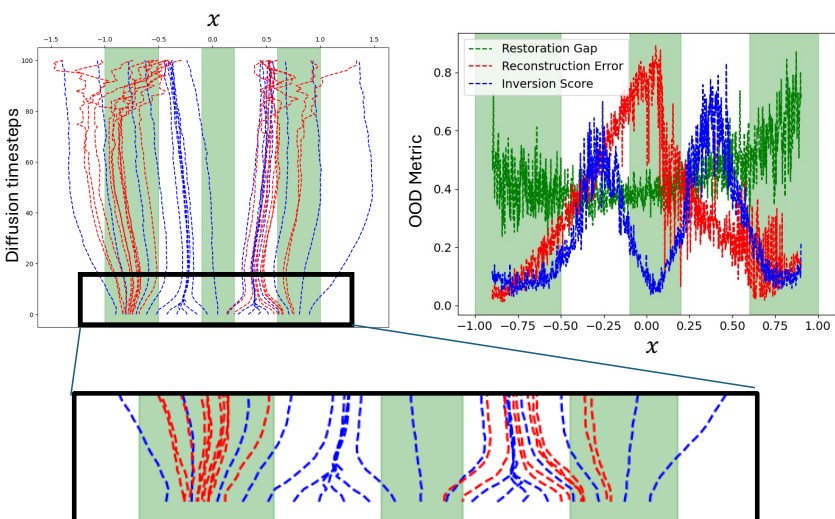

Figure 9: **Comparing DDIM trajectories and associated OOD metrics. Left** shows superimposed DDIM inversion (noising, blue) and denoising (reconstruction, red) paths. The blue lines show samples starting from both ID and OOD regions (e.g., the middle segment) being noised. The red lines show that all trajectories, when denoised, are guided back to the ID (green) regions. **Bottom** highlights the initial steps of the inversion (noising) paths. It illustrates that paths starting from OOD samples exhibit abrupt changes in noising directions, while paths starting in-distribution are smoother. **Right** compares OOD metrics (where a lower score is better). The Inversion Score (blue) accurately identifies the OOD and ID regions. The Reconstruction Error (red) is overly conservative, incorrectly flagging the middle segment as OOD. The Restoration Gap (green) provides no useful signal, failing to distinguish between ID and OOD regions.

# E    ADDITIONAL TAMP SUITE DETAILS

We evaluate our framework on three task domains (`hook reach`, `rearrangement push`, and `rearrangement memory`) with two tasks each. Each of the considered suites focuses on understanding long-horizon success of one particular skill. For example, `hook reach` is about the long-term effect of executing `hook`, while `rearrangement push` focuses on `push` and `rearrangement memory` is designed to confuse the TAMP framework that perform hierarchical planning with non goal-conditioned motion planners. Each task's challenge is directly proportional to the long-horizon action dependency required to complete it. For example, `pull` affects immediately if the next skill is `pick`. But `place` affects the next skill after executing one intermediate skill (like `pick`). Similarly, action dependency is after two skills for `rearrangement push` and `rearrangement memory` tasks. We describe all of such considered tasks below.

1. **Hook Reach (Task 1):**
   - **Scene:** Table with a rack, hook, and cube
   - **Start:** Rack and hook are in workspace, cube is beyond workspace
   - **Goal:** Pick up the cube
   - **Action Skeleton:** pick(hook) → pull(cube, hook) → place(hook) → pick(cube)

2. **Hook Reach (Task 2):**
   - **Scene:** Table with a rack, hook, and cube
   - **Start:** Rack and hook are in workspace, cube is beyond workspace
   - **Goal:** Place the cube on the rack
   - **Action Skeleton:** pick(hook) → pull(cube, hook) → place(hook) → pick(cube) → place(cube, rack)

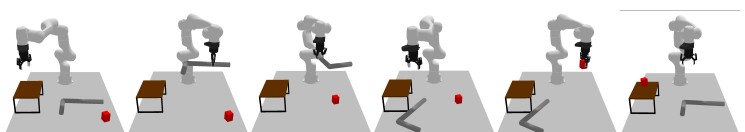

Figure 10: Hook Reach Task 2

3. **Rearrangement Push (Task 1):**
   - **Scene:** Table with a hook, cube, and rack
   - **Start:** Hook and cube are in workspace, rack is beyond workspace
   - **Goal:** Position the cube under the rack
   - **Action Skeleton:** pick(cube) → place(cube) → pick(hook) → push(cube, hook, rack)

4. **Rearrangement Push (Task 2):**
   - **Scene:** Table with a hook, cube, and rack
   - **Start:** Hook is in workspace, cube and rack are beyond workspace
   - **Goal:** Position the cube under the rack
   - **Action Skeleton:** pick(hook) → pull(cube, hook) → place(hook) → pick(cube) → place(cube) → pick(hook) → push(cube, hook, rack)

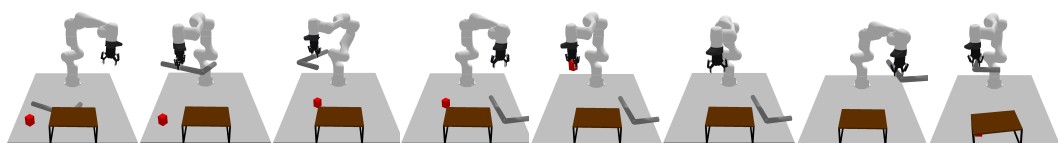

Figure 11: Rearrangement Push Task 2

5. **Rearrangement Memory (Task 1):**

- **Scene:** Table with a hook, red cube, and blue cube
- **Start:** All objects (hook, red cube, blue cube) are in workspace
- **Goal:** Put the red cube where the blue cube is
- **Action Skeleton:** pick(blue_cube) → place(blue_cube) → pick(red_cube) → place(red_cube)

6. **Rearrangement Memory (Task 2):**
   - **Scene:** Table with a hook,red cube, and blue cube
   - **Start:** Hook and blue cube are in workspace, red cube is beyond workspace
   - **Goal:** Put the red cube where the blue cube is
   - **Action Skeleton:** pick(hook) → pull(red_cube, hook) → place(hook) → pick(blue_cube) → place(blue_cube) → pick(red_cube) → place(red_cube)

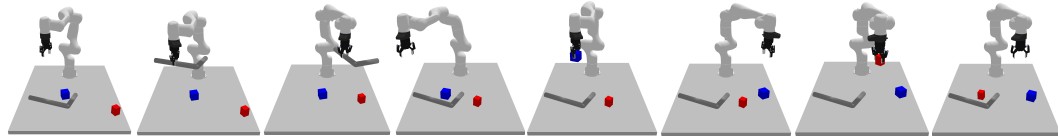

Figure 12: Rearrangement Memory Task 2

## E.1 SKILL STRUCTURE

We consider a finite set of parameterized skills in our skill library. The parameterization, data collection, and training method for each of the skills is described as follows:

1. `Pick`: Gripper picks up an object from the table and the parameters contain 4-DoF pose in the object's frame of reference $(x, y, z, \theta)$.

2. `Place`: Gripper places an object at the target location and parameters contain 4-DoF pose in the place target's frame of reference $(x, y, z, \theta)$. This skill requires specifying two set of parameters, the target pose and the target object (e.g. hook, table).

3. `Push`: Gripper uses the grasped object to push away another object. The skill is motivated from prior work [43, 1] where a hook object is used to `Push` blocks. The parameters of this skill are $(x, y, r, \theta)$ such that the hook is placed at the $(x, y)$ position on the table and pushed by a distance $r$ in the radial direction $\theta$ w.r.t. the origin of the manipulator.

4. `Pull`: Gripper uses the grasped object to pull another object inwards. The skill is also motivated from prior work [43, 1] where a hook object is used to `Pull` blocks. The parameters of this skill are $(x, y, r, \theta)$ such that the hook is placed at the $(x, y)$ position on the table and pulled by a distance $r$ in the radial direction $\theta$ w.r.t. the origin of the manipulator.

## E.2 STATE SPACE OF THE UNIFIED SKILL TRANSITION MODEL

CDGS assumes access to 6D object poses. In practice, we construct the system state as a concatenated vector of poses of objects present in the scenario. We use a fixed object order ([robot, rack, hook, cube1, cube2, . . . ]), passing zero-vectors for absent objects, consistent across all baselines for the experiment.

## F  TRAINING AND SAMPLING: MORE DETAILS ON TAMP EXPERIMENTS

### F.1  CDGS: UNIFIED SCORE MODEL TRAINING

For our TAMP suite, we collect 10000 random skill transition demonstrations for each skill by rolling out random policies in the environment. This ensures enough diversity in the system transitions in the training data. As shown in Fig. 13, we use a mixture-of-experts (MOE) model where we use $N$ feedforward MOE layers. Each layer has a gating network and $M$ experts, where diffusion timestep information is used through an adaptive layer normalization (AdaLN) layer. The outputs from each expert are merged using the predicted gating softmax weights to get the final score of the noisy transition tuple. For OGbnch [48], we just use the datasets provided by them: https://github.com/seohongpark/ogbench

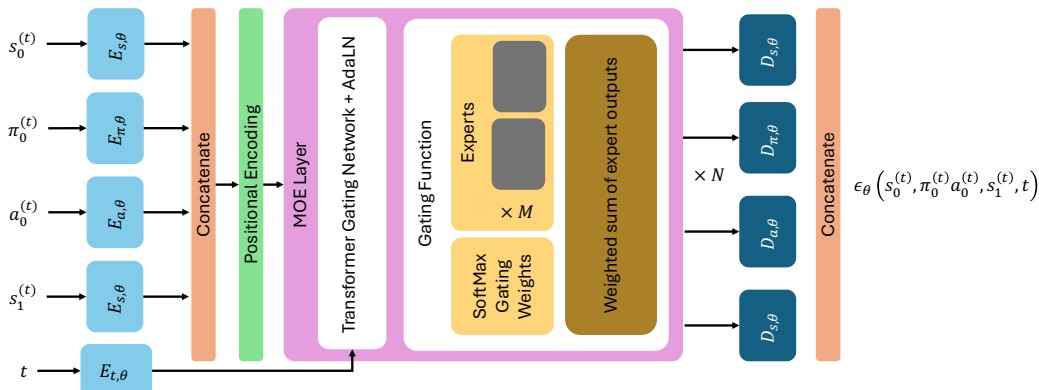

Figure 13: Network architecture for the score function

We particularly use https://huggingface.co/docs/diffusers/en/index library to deploy training and sampling. We provide the training hyperparameters of our setup below:

Table 6: Training setup hyperparameters for CDGS

| Hyperparameter | Value |
| --- | --- |
| Num. MOE layers | 3 |
| Num. Experts per layer | 6 |
| Encoder output dim | 256 |
| Gating network Transformer num. heads | 4 |
| Hidden-dims | 256 |
| Optimizer | torch.AdamW |
| Learning rate | $1e-4$ |
| Positional Encoding | sinusoidal |
| Num. Training Steps | $1e6$ |
| Num Diffusion timesteps | 500 |
| Diffusion $\beta$ schedule | cosine |
| Prediction type | epsilon |

**Effect of training data coverage.** If we consider an "ideal" score function and a perfect representation of the system transition distributions, a solution exists if there is an overlap between the pre-condition and effect of two chosen skills that are required to solve the plan. If such an overlapping segment does not exist, CDGS will not be able to complete the plan. Hence, the training data for each skill must be diverse enough to ensure that the overlap exists. Also, it is worth noting that we use separate dataloaders for all skills to ensure equal distribution of skills in training batches and thus equal preference when sampling.

## F.2 CDGS: Sampling Strategy

For the main denoising loop, we use $T$ denoising timesteps and start with a initial batch size of $B$. For a plan of horizon $H$, we perform the compositional score computation and iterative resampling with $U(t)$ number of resampling steps at every denoising timestep $t$. We devise an adaptive strategy where we apply

1. no pruning for the first few denoising steps until $T_e = k_e T$. We call this as exploration phase. We keep the number of resampling iterations to low during this phase.

2. pruning starts from $T_e = k_e T$ and is done until $T_p = k_p T$. During this, at each denoising timestep, we do some resampling iterations $U(t)$ and then select top-$K$ elites based on the pruning metric.

3. once we have potentially high-quality globally coherent sequences of local modes after a few steps of pruning, we start increasing resampling iterations. This allows us to align the local plans more closely with the optimal mode sequences.

We show the value of each hyperparameter in Tab. 7.

Table 7: TAMP suite experiments: Sampling setup hyperparameters for CDGS

| Hyperparameter | Value |
|---|---|
| Denoising timesteps $T$ | 10 |
| Batch size $B$ | 100 for $H = 7$ and 50 for $H = 4\&5$ |
| Resampling schedule $U(t)$ | $\frac{T - t + 1}{T}(U_T)$ |
| Maximum resampling steps $U_T$ | 50 for $H = 7$ and 40 for $H = 4\&5$ |
| Exploration ends at $k_e$ | 0.7 |
| Pruning ends at $k_p$ | 0.3 |
| Top-$K$ pruning selection | $0.2 \times B$ |
| Pruning objective calculated with $P$ DDIM inversion steps | $0.4 \times T$ |

Thus for a plan of horizon $H$, the total number of function evaluation (NFE) comes to be:

$$NFE = \underbrace{U_T \times \frac{T(T+1)}{2}}_{\text{Main Denoising Loop}} + \underbrace{(k_e - k_p)T \times P}_{\text{Pruning phase}}$$

Since using a single model allows batch operations of converting the $B$ plans of horizon $H$ into a single batched model evaluation with $B \times H$ short transitions.

## F.3 CDGS: Runtime and evaluation

We observe the inference time of CDGS to be $0.5 \times H$ sec (*linear* with $H$) on an Nvidia L40s GPU where $H$ is the plan length. For success metrics, we consider a task success according to the following: (1) **Hook Reach:** the cube is on rack in a stable position (2) **Rearrangement Push:** $\geq 50\%$ of the cube is under the rack and (3) **Rearrangement Memory:** cube within 0.05 m of target positions.

## F.4 STAP [1]

For STAP, we use their policies, critics, and dynamics models trained with their inverse reinforcement learning pipeline (text2motion [36]) available at `https://github.com/agiachris/STAP`. For Rearrangement Push, we modify the criteria of the `Under` predicate such that $>= 50\%$ of the cube must be under the rack to be successful. We train a new model using STAP's code for `Push` and use their pre-trained models for the other skills.

### F.4.1 TASK PLANNING

STAP by itself is only a motion planner. In order to solve full TAMP problems, it must be integrated with an external task planner. Symbolically-feasible skill sequences found by the task planner are evaluated and ranked by STAP for geometric feasibility. For our experiments, we use a BFS-based symbolic planner that searches through a hand-designed PDDL domain.

To better reflect practical considerations, we design the PDDL domain for each task so that provided geometric information is minimized while ensuring that the correct task plan can always be found. We modify the BFS algorithm so that it can revisit previously visited states as the hidden geometric predicates may be different despite the same symbolic predicates.

**Remark on Rearrangement Memory task.** There are two particular characteristics required in a TAMP to solve Rearrangement Memory task:

1. The symbolic planner must understand which particular symbolic state will satisfy the goal condition. Since most required skills are `pick` and `place`, the symbolic effect of all `place` actions are same. As the exact goal position is not embed in the symbolic states, it is not possible for a naïve task planner to solve for a skill sequence.

2. The task planner can give many feasible solutions that the motion planner must evaluate to find the final task and motion plan. This requires goal-conditioned planners. Since STAP uses Q-function based value estimates to evaluate plans, we find that it struggles with the task as it is not a goal-conditioned method.

### F.4.2 CEM SAMPLING

The STAP baselines use a CEM-based sampling algorithm. An initial prior for the actions is sampled for the start state, and then optimized using the value and dynamics models with CEM-optimization. The only difference between Random CEM and STAP CEM is that Random CEM samples the prior from a uniform distribution? (double-check) while STAP samples the prior from its learned policy models

To make a fair comparison between CDGS's diffusion-based sampling and STAP's CEM-based sampling, we match the sampling budget based on the number of function evaluations. Since our unified model serves the same purpose as STAP's policy, value, and dynamics models, we consider evaluating one set of STAP's policy, value, and dynamics models for a skill to be one function evaluation. For STAP, the CEM runs num iterations of sampling for $N * batch_size samples$. Thus, we match the number of sampling iterations and batch size. The exact budgets for each task are given below.

Table 8: CEM-sampling parameters for STAP

| Task | Samples | Iterations | Elites | Total NFE | CDGS NFE |
|------|---------|------------|--------|-----------|----------|
| Hook Reach 1 | 40 | 132 | 16 | 5280 | 2300 |
| Hook Reach 2 | 50 | 165 | 20 | 8250 | 2300 |
| Rearrangement Push 1 | 50 | 165 | 20 | 8250 | 2300 |
| Rearrangement Push 2 | 70 | 336 | 28 | 23520 | 2850 |
| Rearrangement Memory 1 | 40 | 132 | 16 | 5280 | 2300 |
| Rearrangement Memory 2 | 70 | 336 | 28 | 23520 | 2850 |

### F.4.3 UNCERTAINTY QUANTIFICATION

The LLM Planner in text2motion [36] can sometimes generate symbolically invalid actions i.e. (place(cube) when nothing is in hand), which are out-of-distribution for the learned models. text2motion uses a simple ensemble-based OOD detection method (detailed in appendix A.2 of their paper) to filter out symbolically-invalid actions. We use this for all text2motion baselines. For completeness, we also include this in STAP's baselines as **STAP CEM + UQ**, but it does not make any significant improvements.

# G ADDITIONAL ABLATIONS

## G.1 CDGS: A BETTER PRIOR FOR TASK-LEVEL TRAJECTORY SAMPLING

The proposed method constructs a task-level distribution from skill-level distribution given the current state, the intended goal state and the planning horizon. Specifically, CDGS finds a sequence of modes with overlapping pre-condition and effects by systematic exploration and pruning. While this does not always ensure that the plan is symbolically-geometrically feasible, we observe that choosing the top two plans and expanding the BFS tree with system rollouts leads to higher success rates. As shown in Tab. 9, the CDGS (BFS-2) proves to be an upper bound of our approach. This points out that CDGS constructs meaningful task-level distribution with correct task plans.

Table 9: The success rate of the proposed CDGS algorithm is shown and compared with a variant that performs BFS with the top-2 skill chains at every step and uses system dynamics to rollout. All results are calculated from 50 trials for each task.

| | Hook Reach | | Rearrangement Push | | Rearrangement Memory | |
|---|---|---|---|---|---|---|
| | Task 1 | Task 2 | Task 1 | Task 2 | Task 1 | Task 2 |
| Task Length | 4 | 5 | 4 | 7 | 4 | 7 |
| Full Generative TAMP (no PDDL, skill-level data only) | | | | | | |
| CDGS (ours) | 0.64 | 0.58 | 0.84 | 0.48 | 0.42 | 0.18 |
| Full Generative TAMP (no PDDL, skill-level data only) + Rollout with system dynamics | | | | | | |
| CDGS (BFS-2) | 0.72 | 0.64 | 0.90 | 0.62 | 0.48 | 0.22 |

## G.2 ANALYZING SCALING FOR INDIVIDUAL TASKS

We analyze how varying the batch size $B$ and the number of resampling iterations $U$ affects overall planning performance across all long-horizon tasks of horizon ($H$) 4&5 (Hook Reach Task 1

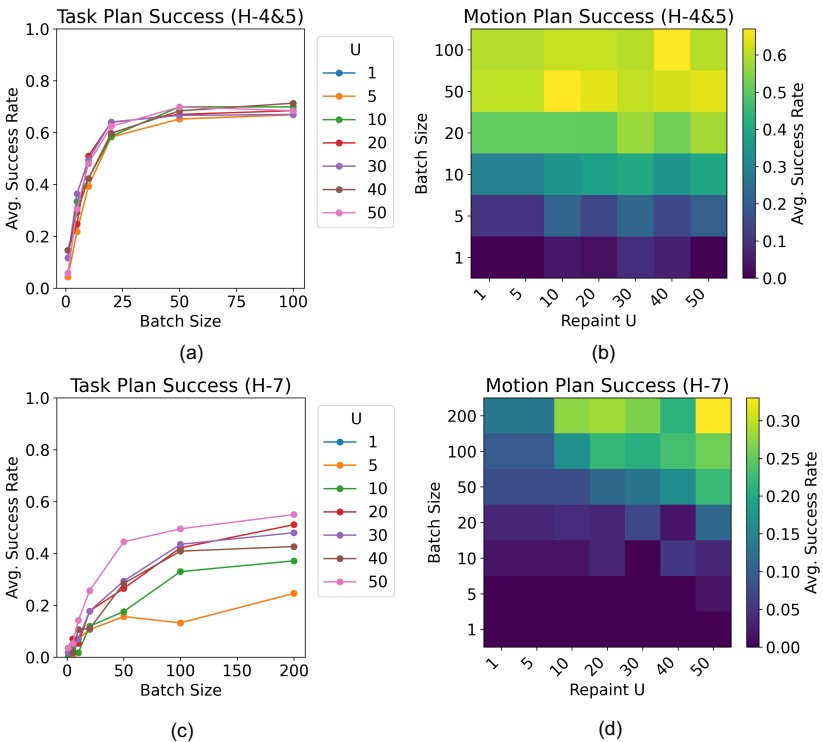

Figure 14: We show the effect of scaling $B$ and $U$ on the overall task planning and motion planning success of CDGS for shorter tasks ($H = 4\&5$) in (a,b) and longer tasks ($H = 7$) in (c,d)

and Task 2; Rearrangement Push Task 1; Rearrangement Memory Task 1) and longer $H = 7$ (Re-

arrangement Push Task 2; Rearrangement Memory Task 2) in Fig. 14. As shown in Fig. 14(a, c), increasing *B* yields a clear, monotonic rise in task-planning success: larger candidate sets diversify the search over the task-level distribution and enable the pruning stage to more reliably identify viable skill sequences. Motion-planning success exhibits a similar trend in Fig. 14(b, d), demonstrating that a more diverse initial sample pool benefits the motion-planning optimization as well. We can also see Fig. 14(b, d) that at lower batch size, increasing the number of resampling steps yields only marginal improvements: without pruning, repeated denoising can still suffer from mode-averaging local minima, where incorrect skill sequences become self-reinforcing. It is only when resampling is coupled with pruning that results in better task planning as well as permit bidirectional "message-passing" of information between the start and goal states—compensating for temporal misalignments at skill (pre-condition and effect) intersection—and thereby unlock significant gains in both task and motion success rates.

## H  HARDWARE SETUP

The experimental setup, illustrated in Fig. 15, consists of the same Franka Panda robot arm, several blocks, a rack, and a hook, observed by an Azure Kinect camera. The camera is mounted in an inclined front-view configuration. AprilTag [57] (https://github.com/fabrizioschiano/apriltag2) markers are used for SE(3) pose detection. We employ Deoxys [70] (https://github.com/UT-Austin-RPL/deoxys_control) for control. After obtaining the SE(3) poses of all the objects: (1) we construct the same environment in simulation, (2) deploy our algorithm in simulation, and (3) execute the planned action in real environment and finally replan.

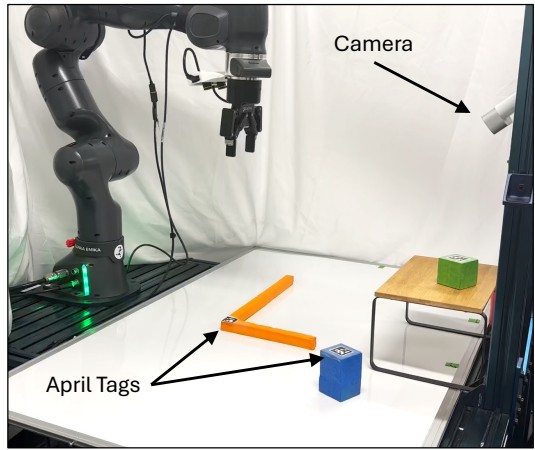

Figure 15: Hardware setup

# I  LLM AND VLM PROMPTING

## I.1  LLM PROMPTING

To make the fairest comparison, we use the same prompt style and in-context examples as text2motion[36], which can be found in Appendix B.2. of their paper. We find that having the model generate multiple candidate task-plans during the shooting phase is critical to task performance, so we add a minimal system prompt to make the LLM instruction-following explicit. An example of a full prompt for **LLM-T2M,** $n = 1$ for `hook Reach` Task 1 is shown below.

---

**User Prompt**

Respond directly in the format specified in the output format section, following the instructions exactly for how many sequences to generate i.e. generate 5 sequences if asked for the "Top 5 robot action sequences".

Available primitives: ['pick(a)', 'place(a, b)', 'pull(a, hook)', 'push(a, hook, rack)']
Available predicates: ['on(a, b)', 'inhand(a)', 'under(a, b)']
Available scene objects: ['table', 'blue_box', 'cyan_box', 'hook', 'rack', 'red_box', 'yellow_box']
Object relationships: ['inhand(hook)', 'on(red_box, rack)', 'on(yellow_box, table)', 'on(blue_box, table)', 'on(cyan_box, rack)', 'on(rack, table)']
Human instruction: could you move all the boxes onto the rack?
Goal predicate set: [['on(yellow_box, rack)', 'on(blue_box, rack)']]
Top 1 robot action sequences: ['pull(yellow_box, hook)', 'place(hook, table)', 'pick(yellow_box)', 'place(yellow_box, rack)', 'pick(blue_box)', 'place(blue_box, rack)']

Available scene objects: ['table', 'rack', 'hook', 'red_box']
Object relationships: ['on(rack, table)', 'on(hook, table)', 'on(red_box, table)']
Human instruction: How would you get the red box in the robot's hand?
Goal predicate set: [['inhand(red_box)']]
Top 5 robot action sequences (python list of lists):

---

**Model Completion**

[ ['pick(red_box)'], ['push(red_box, hook, rack)', 'pick(red_box)'], ['pick(hook)', 'place(hook, table)', 'pick(red_box)'], ['pick(hook)', 'place(hook, rack)', 'pick(red_box)'], ['pick(hook)', 'place(hook, table)', 'push(red_box, hook, rack)', 'pick(red_box)'] ]

---

Interestingly, even though many of the prompts contain partial-to-complete solutions in the in-context examples like the one above, we find that this does not guarantee the LLM will generate the correct plan.

A key piece to text2motion's success for planning despite LLM's lack of geometric awareness is the alternation between shooting and greedy search. When the shooting motion planner fails to find a geometrically feasible motion plan from the 5 task plan candidates, the algorithm falls back to a say-can style greedy search that executes the next action deemed most feasible by a combination of the LLM task planner and the motion planner. This allows the LLM task planner to explore with geometric feedback, and is the reason why text2motion is able to solve `Rearrangement Push` Task 1. For our evaluations, we allow 10 tries, which is much more than the steps required to solve any of the tasks in our evaluation suites.

## I.2 VLM PROMPTING

For VLM experiments, we modify the system prompt and insert scene images before the scene description. Below is an example of `Hook Reach` Task 1 with $n = 11$ in-context examples.

---

**User Prompt**

Respond directly in the format specified in the output format section, following the instructions exactly for how many sequences to generate i.e. generate 5 sequences if asked for the "Top 5 robot action sequences". Review the provided images carefully when constructing your plan.
Available primitives: ['pick(a)', 'place(a, b)', 'pull(a, hook)', 'push(a, hook, rack)']
Available predicates: ['on(a, b)', 'inhand(a)', 'under(a, b)']
Available scene objects: ['table', 'hook', 'rack', 'yellow_box', 'blue_box', 'red_box']
Object relationships: ['inhand(hook)', 'on(yellow_box, table)', 'on(rack, table)', 'on(blue_box, table)']
Human instruction: How would you push two of the boxes to be under the rack?
Goal predicate set: [['under(yellow_box, rack)', 'under(blue_box, rack)'], ['under(blue_box, rack)', 'under(red_box, rack)'], ['under(yellow_box, rack)', 'under(red_box, rack)']]
Top 1 robot action sequences: ['push(yellow_box, hook, rack)', 'push(red_box, hook, rack)']

Available scene objects: ['table', 'blue_box', 'cyan_box', 'hook', 'rack', 'red_box', 'yellow_box']
Object relationships: ['inhand(hook)', 'on(red_box, rack)', 'on(yellow_box, table)', 'on(blue_box, table)', 'on(cyan_box, rack)', 'on(rack, table)']
Human instruction: could you move all the boxes onto the rack?
Goal predicate set: [['on(yellow_box, rack)', 'on(blue_box, rack)']]
Top 1 robot action sequences: ['pull(yellow_box, hook)', 'place(hook, table)', 'pick(yellow_box)', 'place(yellow_box, rack)', 'pick(blue_box)', 'place(blue_box, rack)']

Available scene objects: ['table', 'blue_box', 'hook', 'rack', 'red_box', 'yellow_box']
Object relationships: ['on(hook, table)', 'on(red_box, table)', 'on(blue_box, table)', 'on(yellow_box, rack)', 'on(rack, table)']
Human instruction: Move the ocean colored box to be under the rack and ensure the hook ends up on the table.
Goal predicate set: [['under(blue_box, rack)']]
Top 1 robot action sequences: ['pick(red_box)', 'place(red_box, table)', 'pick(yellow_box)', 'place(yellow_box, rack)', 'pick(hook)', 'push(blue_box, hook, rack)', 'place(hook, table)']

Available scene objects: ['table', 'cyan_box', 'hook', 'red_box', 'yellow_box', 'rack', 'blue_box']
Object relationships: ['on(hook, table)', 'on(red_box, table)', 'on(blue_box, table)', 'on(cyan_box, table)', 'on(rack, table)', 'under(yellow_box, rack)']
Human instruction: How would you get the cyan box under the rack and then ensure the hook is on the table?
Goal predicate set: [['under(cyan_box, rack)', 'on(hook, table)']]
Top 1 robot action sequences: ['pick(blue_box)', 'place(blue_box, table)', 'pick(red_box)', 'place(red_box, table)', 'pick(hook)', 'push(cyan_box, hook, rack)', 'place(hook, table)']

---

Interestingly, we find that including images in the prompt degrades the performance.

---

**User Prompt**

---

Available scene objects: ['table', 'cyan_box', 'hook', 'blue_box', 'rack', 'red_box']
Object relationships: ['on(hook, table)', 'on(rack, table)', 'on(blue_box, table)', 'on(cyan_box, table)', 'on(red_box, table)']
Human instruction: How would you push all the boxes under the rack? Goal predicate set: [['under(blue_box, rack)', 'under(cyan_box, rack)', 'under(red_box, rack)']]
Top 1 robot action sequences: ['pick(blue_box)', 'place(blue_box, table)', 'pick(hook)', 'push(cyan_box, hook, rack)', 'place(hook, table)', 'pick(blue_box)', 'place(blue_box, table)', 'pick(hook)', 'push(blue_box, hook, rack)', 'push(red_box, hook, rack)']

Available scene objects: ['table', 'cyan_box', 'hook', 'rack', 'red_box', 'blue_box']
Object relationships: ['on(hook, table)', 'on(cyan_box, rack)', 'on(rack, table)', 'on(red_box, table)', 'inhand(blue_box)']
Human instruction: How would you set the red box to be the only box on the rack?
Goal predicate set: [['on(red_box, rack)', 'on(blue_box, table)', 'on(cyan_box, table)']]
Top 1 robot action sequences: ['place(blue_box, table)', 'pick(hook)', 'pull(red_box, hook)', 'place(hook, table)', 'pick(red_box)', 'place(red_box, rack)', 'pick(cyan_box)', 'place(cyan_box, table)']

Available scene objects: ['table', 'cyan_box', 'red_box', 'hook', 'rack']
Object relationships: ['on(hook, table)', 'on(rack, table)', 'on(cyan_box, rack)', 'on(red_box, rack)']
Human instruction: put the hook on the rack and stack the cyan box above the rack - thanks
Goal predicate set: [['on(hook, rack)', 'on(cyan_box, rack)']]
Top 1 robot action sequences: ['pick(hook)', 'pull(cyan_box, hook)', 'place(hook, rack)', 'pick(cyan_box)', 'place(cyan_box, rack)']

Available scene objects: ['table', 'cyan_box', 'hook', 'rack', 'red_box', 'blue_box']
Object relationships: ['on(hook, table)', 'on(blue_box, rack)', 'on(cyan_box, table)', 'on(red_box, table)', 'on(rack, table)']
Human instruction: Move the warm colored box to be underneath the rack.
Goal predicate set: [['under(red_box, rack)']]
Top 1 robot action sequences: ['pick(blue_box)', 'place(blue_box, table)', 'pick(red_box)', 'place(red_box, table)', 'pick(hook)', 'push(red_box, hook, rack)']

Available scene objects: ['table', 'blue_box', 'red_box', 'hook', 'rack', 'yellow_box']
Object relationships: ['on(hook, table)', 'on(blue_box, table)', 'on(rack, table)', 'on(red_box, table)', 'on(yellow_box, table)']
Human instruction: situate an odd number greater than 1 of the boxes above the rack
Goal predicate set: [['on(blue_box, rack)', 'on(red_box, rack)', 'on(yellow_box, rack)']]
Top 1 robot action sequences: ['pick(hook)', 'pull(blue_box, hook)', 'place(hook, table)', 'pick(blue_box)', 'place(blue_box, rack)', 'pick(red_box)', 'place(red_box, rack)', 'pick(yellow_box)', 'place(yellow_box, rack)']

**User Prompt**

continued...

Available scene objects: ['table', 'cyan_box', 'hook', 'yellow_box', 'blue_box', 'rack']
Object relationships: ['on(hook, table)', 'on(yellow_box, rack)', 'on(rack, table)', 'on(cyan_box, rack)']
Human instruction: set the hook on the rack and stack the yellow box onto the table and set the cyan box on the rack
Goal predicate set: [['on(hook, rack)', 'on(yellow_box, table)', 'on(cyan_box, rack)']]
Top 1 robot action sequences: ['pick(yellow_box)', 'place(yellow_box, table)', 'pick(hook)', 'pull(yellow_box, hook)', 'place(hook, table)']

Available scene objects: ['table', 'rack', 'hook', 'cyan_box', 'yellow_box', 'red_box']
Object relationships: ['on(yellow_box, table)', 'on(rack, table)', 'on(cyan_box, table)', 'on(hook, table)', 'on(red_box, rack)']
Human instruction: Pick up any box.
Goal predicate set: [['inhand(yellow_box)'], ['inhand(cyan_box)']] Top 1 robot action sequences: ['pick(yellow_box)']

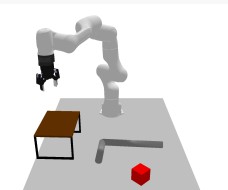 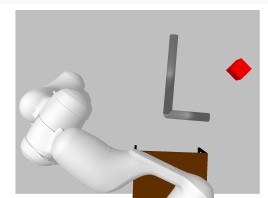 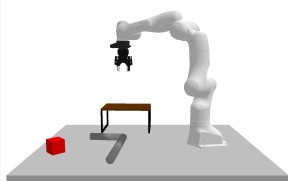

Available scene objects: ['table', 'rack', 'hook', 'red_box']
Object relationships: ['on(rack, table)', 'on(hook, table)', 'on(red_box, table)']
Human instruction: How would you get the red box in the robot's hand? Goal predicate set: [['inhand(red_box)']]
Top 5 robot action sequences (python list of lists):

**Model Completion**

[ ['pick(red_box)'], ['pick(hook)', 'place(hook, table)', 'pick(red_box)'], ['pick(rack)', 'place(rack, table)', 'pick(red_box)'], ['pick(hook)', 'place(hook, table)', 'pick(rack)', 'place(rack, table)', 'pick(red_box)'], ['pick(red_box)', 'place(red_box, table)', 'pick(red_box)'] ]

## J  PSEUDO-CODE AND HYPERPARAMETERS FOR IMAGE GENERATION

```python
@torch.no_grad()
def text2panorama_noise_resample_pruning(self, prompts,
    height=512, width=2048, num_inference_steps=50,
    guidance_scale=7.5, num_samples_per_prompt=10, top_K=0.2
):

    # Prompts -> text embeds
    text_embeds = self.get_text_embeds(prompts)

    # Define panorama grid and get views for individual segments
    views, covered_width = get_views_gtamp(height, width)
    latent = torch.randn((
        num_samples_per_prompt,
        self.unet.in_channels,
        height // 8,
        covered_width
    ), device=self.device)
    count = torch.zeros_like(latent)
    value = torch.zeros_like(latent)

    self.scheduler.set_timesteps(num_inference_steps)

    with torch.autocast('cuda'):
        num_timesteps = len(self.scheduler.timesteps)
        for i, t in enumerate(tqdm(self.scheduler.timesteps)):
            U = int(
                min(
                    max(
                        (float(i) / float(len(self.scheduler.timesteps))) * \
                        self.num_resampling_steps,
                        5
                    ),
                    self.num_resampling_steps
                )
            )
            for u in tqdm(range(U), leave=False):
                count.zero_()
                value.zero_()

                all_latents = []

                for h_start, h_end, w_start, w_end in views:
                    latent_view = latent[:, :, h_start:h_end, w_start:w_end]
                    all_latents.append(latent_view)

                latent_view = torch.stack(all_latents, dim=0) # [N, B, C, H, W]
                N, B = latent_view.shape[0], latent_view.shape[1]
                latent_view_batched = latent_view.view(
                    -1,
                    *latent_view.shape[2:]
                ) # [N*B, C, H, W]

                positive_text_embeds, negative_text_embeds = text_embeds.chunk(2)
                positive_text_embeds = positive_text_embeds.repeat(N*B, 1, 1)
                negative_text_embeds = negative_text_embeds.repeat(N*B, 1, 1)
                text_embeds_batched = torch.cat([
                    positive_text_embeds,
                    negative_text_embeds
                ], dim=0) # [2*N*B, 77, 768]

                latent_model_input = torch.cat([latent_view_batched] * 2, dim=0)

                noise_pred = self.unet(
```

```python
            latent_model_input,
            t,
            encoder_hidden_states=text_embeds_batched)['sample']

        # perform guidance
        noise_pred_uncond, noise_pred_cond = noise_pred.chunk(2)
        noise_pred = noise_pred_uncond + \
            guidance_scale * (noise_pred_cond - noise_pred_uncond)

        noise_pred_batched = noise_pred.view(
            N, B,
            noise_pred.shape[-3],
            noise_pred.shape[-2], noise_pred.shape[-1]
        ) # [N, B, C, H, W]

        for idx, (h_start, h_end, w_start, w_end) in enumerate(views):
            noise_pred_batched_view = noise_pred_batched[idx] # [B, C,
            # compute the denoising step with the reference model
            value[:, :, h_start:h_end, w_start:w_end] += noise_pred_batched_view
            count[:, :, h_start:h_end, w_start:w_end] += 1

        noise_combined = torch.where(count > 0, value / count, value)
        latent = self.scheduler.step(noise_combined, t, latent)

        if u < U-1 and i < len(self.scheduler.timesteps)-1 and i > 0:
            pred_x0 = latent['pred_original_sample']
            latent = latent['prev_sample']
            latent = self.undo_step(latent, pred_x0, noise_combined, t)
        elif u == U-1 and \
            (i < 0.4*num_timesteps and i > 0.1*num_timesteps):
            pred_x0 = latent['pred_original_sample']
            latent = latent['prev_sample']
            latent = self.inversion_pruning(
                pred_x0,
                latent,
                text_embeds_batched,
                views,
                guidance_scale,
                top_K
            )
        else:
            latent = latent['prev_sample']

return latent
```

```python
def inversion_pruning(self, pred_x0, latents, text_embeds, views,
    guidance_scale, top_K
):

    num_models = len(views)
    B = pred_x0.shape[0]
    all_timesteps = self.scheduler.timesteps.flip(dims=(0,))
    num_inference_steps = len(all_timesteps)

    batched_x0s = []
    for h_start, h_end, w_start, w_end in views:
        batched_x0s.append(pred_x0[:, :, h_start:h_end, w_start:w_end])

    batched_x0s = torch.stack(batched_x0s, dim=0)  # [num_models, N, C, H, W]
    batched_x0s = batched_x0s.view(
        num_models * B, -1,
        batched_x0s.shape[-2],
        batched_x0s.shape[-1]
    )

    inversion_latents = batched_x0s.clone()
    all_noise_prediction = []

    for idx, i in tqdm(
        enumerate(all_timesteps[:-num_inference_steps//2+1]),
        leave=False,
        total=num_inference_steps-1
    ):
        t = i
        t_next = all_timesteps[idx + 1]
        alpha_t = self.scheduler.alphas_cumprod[t]
        alpha_t_next = self.scheduler.alphas_cumprod[t_next]
        sqrt_alpha_t = torch.sqrt(alpha_t)
        sqrt_alpha_t_next = torch.sqrt(alpha_t_next)
        sqrt_one_minus_alpha_t = torch.sqrt(1 - alpha_t)
        sqrt_one_minus_alpha_t_next = torch.sqrt(1 - alpha_t_next)

        with torch.no_grad():
            latent_model_input = torch.cat([inversion_latents] * 2)
            noise_pred = self.unet(
                latent_model_input,
                t,
                encoder_hidden_states=text_embeds
            )['sample']
            noise_pred_uncond, noise_pred_cond = noise_pred.chunk(2)
            noise_pred_combined = noise_pred_uncond + \
                guidance_scale * (noise_pred_cond - noise_pred_uncond)

            x0_pred = (inversion_latents - \
                sqrt_one_minus_alpha_t * noise_pred_combined) / sqrt_alpha_t
            x0_pred = torch.clamp(x0_pred, -1.0, 1.0)
            noise_pred_combined = (inversion_latents - \
                sqrt_alpha_t * x0_pred) \
                / sqrt_one_minus_alpha_t
            inversion_latents = sqrt_alpha_t_next * x0_pred + \
                sqrt_one_minus_alpha_t_next * noise_pred_combined
            all_noise_prediction.append(noise_pred_combined)

    all_intermediate_noise_preds = torch.stack(all_noise_prediction, dim=1)
    derivative = torch.diff(all_intermediate_noise_preds, dim=1)

    all_scores = torch.norm(
        derivative.reshape(num_models*B, -1),
        dim=1
    ).reshape(num_models, B)
```

```
final_scores = all_scores.mean(dim=0) # (B,)

num_selected_samples = max(int(top_K * B), 1)
topk_indices = torch.topk(
    final_scores,
    k = num_selected_samples,
    largest=False
)[1]

arranged_batch = latents.clone()
arranged_batch = arranged_batch[topk_indices]

while arranged_batch.shape[0] < B:
    arranged_batch = torch.cat([arranged_batch, arranged_batch], dim=0)

arranged_batch = arranged_batch[:B]

return arranged_batch
```

Table 10: Sampling setup hyperparameters for panorama generation experiments

| Hyperparameter | Value |
|---|---|
| Denoising timesteps $T$ | 50 |
| Batch size $B$ | 10 |
| Composition weights $\gamma_{1:H}$ | 0.5 |
| Resampling schedule $U(t)$ | $\dfrac{T-t+1}{T}(U_T)$ |
| Maximum resampling steps $U_T$ | 10 |
| Exploration ends at $k_e$ | 0.2 |
| Pruning ends at $k_p$ | 0.5 |
| Top-$K$ pruning selection | $0.4 \times B$ |
| Pruning objective calculated with $P$ DDIM inversion steps | $0.5 \times T$ |

All experiments were run on single NVIDIA™ L40s or NVIDIA™ A100 GPUs.

## K  PSEUDO-CODE AND HYPERPARAMETERS FOR VIDEO GENERATION

We modify CogVideoX pipeline provided in Huggingface: https://github.com/huggingface/diffusers/blob/v0.35.1/src/diffusers/pipelines/cogvideo/pipeline_cogvideox.py.

We keep the logic same as images.

Table 11: Sampling setup hyperparameters for long-video generation experiments

| Hyperparameter | Value |
|---|---|
| Denoising timesteps $T$ | 30 |
| Batch size $B$ | 10 |
| Composition weights $\gamma_{1:H}$ | 0.5 |
| Resampling schedule $U(t)$ | $\dfrac{T-t+1}{T}(U_T)$ |
| Maximum resampling steps $U_T$ | 10 |
| Exploration ends at $k_e$ | 0.3 |
| Pruning ends at $k_p$ | 0.6 |
| Top-$K$ pruning selection | $0.4 \times B$ |
| Pruning objective calculated with $P$ DDIM inversion steps | $0.5 \times T$ |

All experiments were run on single NVIDIA™ H100 GPUs.

# L    SCALING ANALYSIS: NFE AND WALL CLOCK TIMES

In general, we consider $T$ denoising iterations for which we perform $U$ steps of iterative resampling to get the candidate global plans and then perform $T$ steps of DDIM inversion steps to prune infeasible candidates. Eventually, at each denoising step, CDGS selects the best-K denoising paths. We repeat the selected denoising paths to fill up the batch for the next denoising step. Since we use stochasticity in the main denoising loop, the same denoising paths can lead to different clean samples.

Thus, we can compute the NFEs as:

$$NFE = T \times U + T \times T \tag{6}$$

If we consider model inference complexity to be $O(1)$, the computational complexity of CDGS is $O(T^2)$ if $T \geq U$ else it is $O(U^2)$. To give a comparison CDGS is $(U + T)$ times more expensive to run than naïve compositional sampling.

To reduce the complexity and compute requirements, we perform some engineering-modifications:

1. we observe that early pruning does not help a lot since Tweedie estimates for noisy samples at higher noise levels are not very accurate, hence:
   (a) instead of always performing $U$ resampling steps, we gradually increase $U$ throughout the denoising process such that we do not overfit to bad denoising paths at earlier timesteps
   (b) we can deploy pruning only for the last 20% denoising iterations
   
   this makes effective number of resampling steps approximately $U/2$.
2. we also observe that abrupt direction and magnitude changes of score functions are more prominent in the initial DDIM inversion steps (eventually it stabilizes as noisy latents come in-distribution), allowing us to stop DDIM inversion steps at $T/2$.

This allows making CDGS only $(0.5U + 0.1T)$ more expensive than naïve compositional diffusion. To give a practical example, by incorporating jit compilation, a single model inference for Stable Diffusion 2.1 takes 1.5 secs on a NVIDIA™ L40s GPU and with $T = 50$ it takes 75 secs to generate panoramic image using naïve compositional sampling. With $U = 10$ and pruning happening for $0.2T$ steps, with CDGS it takes around 700 secs.

Compute and wall-clock time are completely dependent on the base local generative model and the number of inference steps required to generate a good sample from it. For example we observe that for toy and robotics domains, $T = 50$ is sufficient to sample good solutions. Also, note that, for a batch of $B$ candidate global plan for horizon $H$ each with $M$ local segments, we construct a batch of local segments of size $B \times M$ to denoise all the local segments in parallel for every denoising step. This step depends on the available GPU memory, which limits the maximum batch size.

## L.1    TOY DOMAIN

We analyze the runtime and success of scaling inference-time compute in the toy-domain. All experiments in this section were run on single NVIDIA™ V100 GPUs.

In our first experiment, we disable pruning and ablate the number of resampling steps $U$ as shown in Fig. 16. We find that:

1. wall-clock time scales linearly with the additional compute
2. increasing resampling steps can address declining performance as horizons increase
3. overall, a key finding is that the improvement in performance with increasing $U$ diminishes as the horizon increases

In our second experiment, we ablate the choice of the parameters for pruning: start and end. We find that:

1. the cost of pruning increases wall-clock linearly.

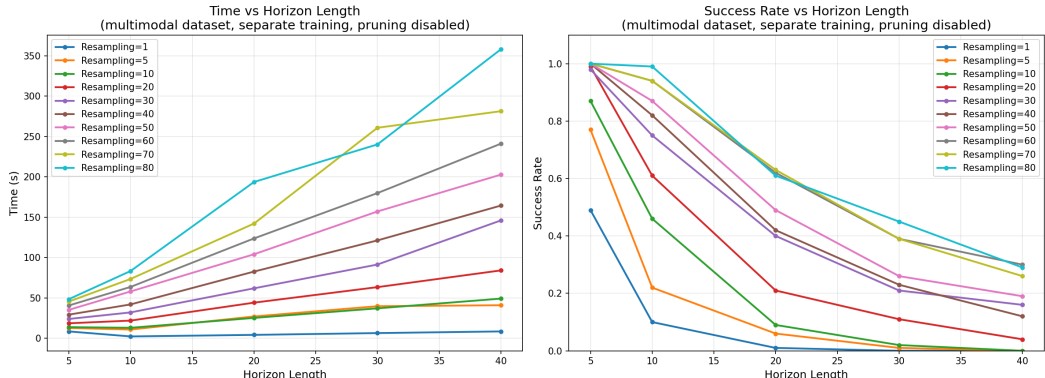

Figure 16: Left: runtime scales linearly with horizon length and resampling steps. Right: success rates decline over horizon lengths, but this is alleviated by additional resampling.

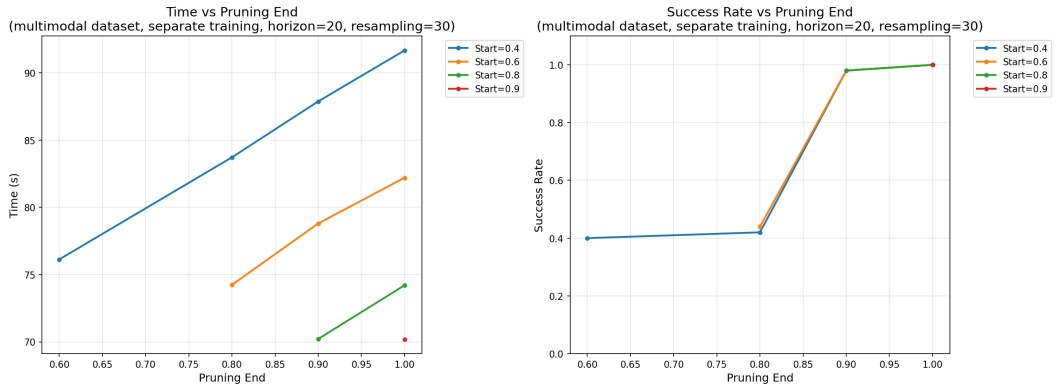

Figure 17: Comparison of time and success heatmaps for pruning

2. we perform this experiment with horizon $H = 20$ and number of resampling steps $U = 30$. Bu adding pruning, we note that with minimal increase in wall-clock time (around 5%), we can push the success rate to be 100%.

3. One additional insight we obtained is that pruning until the end of the denoising process is essential. This supports our key insight that as Tweedie estimates get accurate at lower noise levels, pruning becomes more effective in selecting better denoising paths.

It is worth noting that because of independently sampling local segments, the complexity of the problem increases exponentially with horizon. For example, for a horizon of $H = 5$ and each transition having two feasible modes, there can be $2^H$ possible sequences of feasible factor modes; only two of them will be valid for coherent global plan synthesis. **CDGS is able to navigate this exponentially increasing domain by linearly scaling the compute and memory requirements.**

## L.2 OGBENCH DOMAIN

We report the wall clock times and associated gain in performance for the OGbench Maze domains (similar for both PointMaze and AntMaze) in Fig. 18. It is worth noting that CDGS uses more compute to scale performance even with naïve compositional methods.

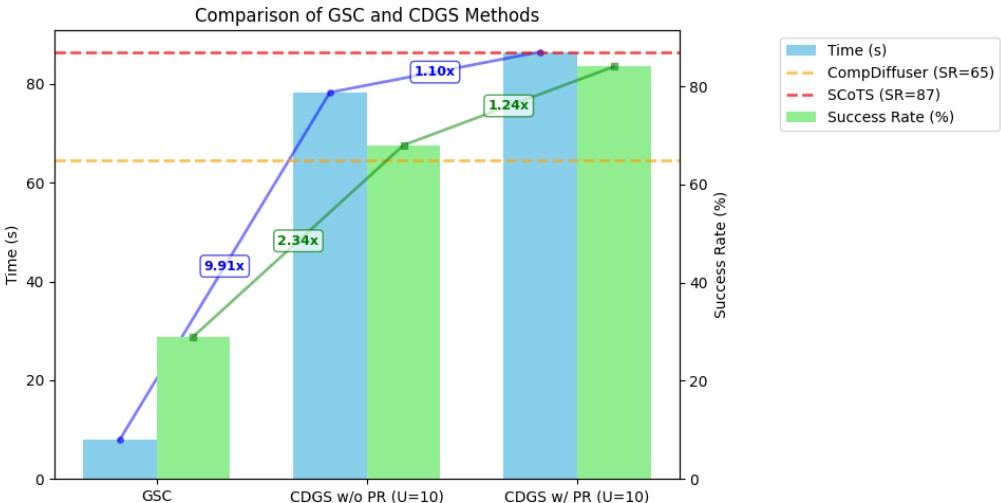

Figure 18: We show results for OGBench maze tasks. We observe that performance improves with adding resampling steps along with additional computational time. With pruning, we see more improvement in performance with only an additional 10% compute time. Overall, CDGS with resampling and pruning takes around 10-12x more time than GSC. This relationship validates that CDGS scales linearly with number of resampling steps and pruning.

It should be noted that CDGS with resampling and pruning can scale the performance of naïve compositional sampling, in a training-free manner, to an extent that:

1. beats baselines like CompDiffuser [40] that use overlap information while training and learn an overlap conditioned score function.

2. performs on par with baselines like SCoTS [32] that use data augmentation to synthesize datasets with long-horizon data and train a policy on the new dataset.

# M COMPOSITIONAL DIFFUSION WITH GUIDED SEARCH: COMPLETE ALGORITHM FOR MOTION PLANNING

**Problem with the TAMP benchmark:** TAMP benchmark is based on skill-level action learning. For example, for `push` skill, this means that instead of learning the low-level end-effector motion, we are learning start pose of the end effector wrt the target object's position (here, cube) and by how much we want to gripper to move to complete the skill execution. This structure implies that for every skill, only certain objects can move in the environment, and the other objects remain static.

**Subproblem 1: What happens when more objects move in the predicted state of the planner?** Since the planner is trained on diverse set of skill transitions and predicts the sequence of $\{(s_i, \pi_i, a_i, s_{i+1})\}$, it is likely that for a particular predicted skill $\pi_i$, for example `pull`, objects other than the target cube move in the predicted next state of the transition. For DDIM inversion objective, even if the planned transition of the target object is correct, it will reject the transition as other objects have moved too.

**Solution:** We use learned forward dynamics model per skill to ensure that only the objects relevant to the predicted skill move for a planned transition. Basically for every predicted skill in the planned sequence of CDGS $\{(s_i, \pi_i, a_i, s_{i+1})\}$, we use forward dynamics model $f_{\pi_i}$ to overwrite $s_{i+1} = f_{\pi_i}(s_i, a_i)$ such that only the pose of target objects (hook, gripper and target cube in case of `pull` skill) to change and other objects remain static. This allows DDIM inversion to evaluate and score planned local transitions appropriately.

**Changes in algorithm to incorporate the solution:**

---

**Algorithm 3** CDGS

---

**Require:** Start $x_s$, Goal $x_g$, Planning horizon $H$
**Require:** Diffusion noise schedule,
**Require:** Pretrained local plan score function $\varepsilon_\theta(y^{(t)}, t)$,
**Require:** number of candidate plans $B$, number of elite plans $K$ at every step
1: Initialize $B$ global plan candidates: $\tau^{(T)}$
2: $\tau^{(T)} = (y_1^{(T)} \circ \cdots \circ y_M^{(T)}) \sim \mathcal{N}(0, \mathbf{I})$
3: **for** $t = T, \ldots, 1$ **do**
4: $\quad \varepsilon(\tau^{(t)}, t) = \text{ComposedScore}(\tau^{(t)}, t, \varepsilon_\theta, x_s, x_g)$
5: $\quad \hat{\tau}_0^{(t)} = (\tau^{(t)} - \sqrt{1 - \alpha_t}\varepsilon(\tau^{(t)}, t))/\sqrt{\alpha_t}$
6: $\quad \hat{\tau}_{0,new}^{(t)} = \text{LearnedForwardDynamics}(\hat{\tau}_0^{(t)})$
7: $\quad$ Rank plans using $J(\hat{\tau}_{0,new}^{(t)})$ Eq. 5
8: $\quad$ Select best-$K$ global plans
9: $\quad$ Repopulate candidates using filtered plans
10: $\quad \tau^{(t-1)} \sim p(\tau^{(t-1)} | \tau^{(t)}, \hat{\tau}_0^{(t)})$ Eq. 2
11: **end for**
12: return $\tau^{(0)}$

---