# OpenReview forum: "Compositional Diffusion with Guided search for Long-Horizon Planning"
_ICLR.cc/2026/Conference — ICLR 2026 Oral_

### Official Review · Reviewer_xJxE · 2025-10-18

**Soundness:** 3
**Presentation:** 3
**Contribution:** 3
**Rating:** 6
**Confidence:** 4

**Summary:**

The author introduces Compositional Diffusion with Guided Search (CDGS), a novel framework for generating coherent long-horizon plans and content by addressing the critical challenge of mode-averaging in compositional generative models. Existing compositional approaches, while efficient in leveraging short-horizon local generative models, often average incompatible modes when local distributions are multimodal, leading to globally incoherent or locally infeasible outputs. CDGS integrates a guided search mechanism directly into the diffusion denoising process to overcome this limitation. The authors demonstrate the effectiveness of CDGS across three challenging long-horizon domains: robotic manipulation planning, panoramic image generation, and long video synthesis, showing significant improvements over naive compositional baselines.

**Strengths:**

1. The paper is well-structed and easy to follow. The proposed approach is well-grounded in motivation, and the empirical results provide compelling evidence of its effectiveness.
2. The paper tackles a fundamental and highly significant problem in generative modeling.

**Weaknesses:**

1. Some implementation details are missing to fully replicate the method. Notably, the candidate "repopulation" step lacks clarity on how new batches are generated from elite plans.
2. CDGS requires additional resampling iterations and pruning steps which can require many additional forward passes of the model. But the paper doesn’t report the wall-clock runtime or compute comparisons, which makes it hard to judge the practical overhead.
3. Although this paper focuses on inference-time trajectory stitching methods, I suggest the authors include related works on trajectory data augmentation methods (e.g., [1,2]) that address similar problems.
4. CDGS uses a curvature-based approach to approximate likelihoods for pruning low-quality plans. However, there are existing reconstruction-based methods for evaluating sample quality in diffusion models, such as restoration gap [3] and minority score [4]. Could you clarify the advantages of the curvature-based approach over reconstruction-based alternatives for likelihood approximation?
5. (optional) It would strengthen the paper if the authors could provide empirical comparisons demonstrating the effectiveness of the curvature-based metric compared to reconstruction-based approaches for plan pruning.

[1] State-Covering Trajectory Stitching for Diffusion Planners, 2025

[2] Extendable Long-Horizon Planning via Hierarchical Multiscale Diffusion, 2025

[3] Refining Diffusion Planner for Reliable Behavior Synthesis by Automatic Detection of Infeasible Plans, 2023

[4] Don’t Play Favorites: Minority Guidance for Diffusion Models, 2024

Note: I still think this work is promising. Although a borderline accept is given currently, I'd be open to increasing my score upon seeing suitable revisions or clarification addressing my concerns.

**Questions:**

see weaknesses

---

> ### Author Response · Authors · 2025-11-19
> **Rebuttal by Authors**
>
> Thank you for your valuable feedback. Please find our explanations below.
>
> **W1. the candidate "repopulation" step lacks clarity on how new batches are generated from elite plans.**
>
> Thanks for pointing this out. At each denoising step, CDGS selects the best-K denoising paths. We repeat the selected denoising paths to fill up the batch for the next timestep. Since we use stochasticity in the main denoising loop, the same denoising paths can lead to different clean samples. We have added this clarification in the implementation details in appendix M.
>
> **W2. Runtime analysis**
>
> We have provided a detailed runtime analysis in the Appendix M, please see the revised paper. In summary, CDGS incurs a predictable linear overhead relative to na\"ive compositional diffusion, and in practice the additional cost scales up the performance significantly as well (the core essence of inference-time scaling). Across domains, we observe that CDGS is $(0.5U + 0.1T)$x more expensive to run where $U$ is the number of resampling steps and $T$ is the number of denoising iterations. In most of our experiments, we conclude that CDGS typically requires 10–12x more compute than a baseline compositional sampler when both resampling and pruning are enabled (across OGBench and panorama synthesis). The exact wall clock time depends on the base local generative model and the number of inference steps required to generate a good sample from it. Engineering optimizations—batching local segments, limiting pruning to low-noise steps, and JIT compilation—keep the wall-clock scaling low. Importantly, this computational cost consistently yields substantial improvements in global coherence across all evaluated domains.
>
> It is worth noting that because of independently sampling local segments, the complexity of the problem increases exponentially with horizon. For a horizon of $H$ and each local segment having $Z$ feasible modes, there can be $Z^H$ possible sequences of feasible global plan modes, only a few of them will be valid for coherent global plan synthesis. CDGS is able to navigate this exponentially increasing domain by linearly scaling the compute and memory requirements.
>
> **W3 include related works on trajectory data augmentation methods**
>
> We have added the relevant literature in the related works section.
>
> **W4 and W4: Could you clarify the advantages of the curvature-based approach over reconstruction-based alternatives for likelihood approximation?**
>
> We have provided an empirical comparison of the different pruning methods in Appendix E.1. In summary we see two directions of improvement when using CDGS's curvature based metric vs other reconstruction-based alternatives:
> - DDIM inversion only requires forward noising while reconstruction methods require both forward noising and denoising back (for multiple choices of diffusion timestep).
> - For distributions with disjoint modes (like the one considered for our 1D experiment), it is not necessary that the reconstructed sample after noising and denoising will belong to the same mode as the original clean sample. This makes reconstruction based metrics invalid or overly conservative neglecting in-distribution segments.
>
>
> Thanks again. We will be happy to answer any other follow up questions.

---

> > ### Comment · Reviewer_xJxE · 2025-11-20
> >
> > I thank the authors for the detailed clarifications. All of my concerns have been well addressed, and I raised my score accordingly. I enjoyed reading this work. Good luck!

---

### Official Review · Reviewer_Qy6u · 2025-10-28

**Soundness:** 3
**Presentation:** 3
**Contribution:** 3
**Rating:** 8
**Confidence:** 4

**Summary:**

In this paper, the authors propose Compositional Diffusion with Guided Search (CDGS), a novel framework for compositional generation using diffusion models. The work primarily addresses the "mode-averaging" problem inherent in prior score-averaging approaches. This issue arises when composing multi-modal local distributions, where existing methods tend to average incompatible local modes, resulting in globally incoherent and infeasible plans.

To overcome this limitation, CDGS introduces two key inference-time strategies. First, it employs iterative resampling of local generations based on the compositional score. This process functions as an effective message-passing mechanism to propose globally coherent candidate samples by propagating information across the entire sequence. Second, the framework filters these candidates using a likelihood-based pruning strategy. It generates a diverse population of candidate plans and then selects and re-populates the top-*K* candidates based on a proposed local feasibility metric. The authors offer a novel approach to approximate this metric by leveraging the insight that high-likelihood samples follow low-curvature paths during the DDIM inversion process. Specifically, this curvature is measured by the norm of the time-derivative of the score network's noise prediction, allowing for an efficient, likelihood-informed search.

The authors empirically validate CDGS across a diverse set of long-horizon tasks, including robotic planning, panoramic image generation, and long video generation. The results consistently demonstrate the superiority of CDGS over previous methods. The paper further substantiates the efficacy of its components through ablation studies and detailed analyses on scaling effects. Notably, the method demonstrates strong performance on complex robotic planning tasks where baseline approaches like GSC fail. Even when provided with the oracle symbolic task plan, GSC struggles with compositional motion planning due to mode-averaging, whereas CDGS successfully finds a coherent solution by integrating task and motion planning within its guided search, highlighting the critical importance of the proposed mechanism. The authors ensure reproducibility by providing code and detailed experimental settings in the appendix, which significantly aids in the concrete understanding of the proposed method.

**Strengths:**

- **Clear Problem Formulation and an Effective Solution**: The paper clearly articulates critical limitations of prior score-averaging based compositional diffusion methods—the mode-averaging and global incoherence problems. It then introduces a well-motivated and effective solution, CDGS, which synergistically combines two key strategies: (1) iterative resampling to enforce global coherence through message passing , and (2) a novel likelihood-based pruning mechanism to filter out locally infeasible candidates. This problem-solution framing is a significant strength of the paper.

- **Convincing Empirical Validation across Diverse Domains**: The generalizability of CDGS is convincingly demonstrated through extensive experiments on a variety of challenging long-horizon tasks, including robotic planning , panoramic image generation , and long video generation. The consistent and significant performance improvements over baselines across these different domains strongly support the claim that CDGS is a broadly applicable, task-agnostic framework.

- **Thoroughness and Commitment**: The paper is well-supported and transparent, which significantly strengthens its contribution.

    - **Reproducibility**: The authors provide a link to the source code and meticulously detail all hyperparameters for the different experimental domains in the appendices (e.g.,  Appendices G, K, L) .

    - **In-depth Analysis and Discussion**: The appendices offer deep dives into crucial components, providing a detailed derivation of the pruning objective (Appendix E) , a thoughtful comparison with prior methods (Appendix D) , and insightful ablation studies on scaling effects (Appendix H).

    - **Transparency**: The authors transparently document the full experimental setup, including prompts and task specifications (Appendices C, F, I, J) , and proactively discuss the limitations of their work (Appendix A), demonstrating a mature and thoughtful research approach.

**Weaknesses:**

- **Lack of Adaptive Computation and Backtracking**: While CDGS effectively overcomes limitations of prior methods through inference-time search, it lacks adaptability.
	- **Fixed Computational Overhead**: The proposed method utilizes a fixed computational budget regardless of task difficulty, as determined by hyperparameters (Appendix G.2) . This can lead to inefficient, excessive computation for simpler tasks, or insufficient computation for more complex ones.
	- **Absence of Backtracking**: CDGS does not support backtracking, which limits its ability to explore diverse possibilities at various depths—a crucial capability for solving complex reasoning and generation problems. In contrast, several recent inference-time scalable diffusion models [1,2,3] offer both adaptability to solve problems within a given budget and backtracking for effective deep search.

- **Ambiguous Link Between Iterative Resampling and Mode-Averaging**: The paper presents iterative resampling and likelihood-based pruning as solutions to the mode-averaging problem. While both contribute to this goal, their primary roles appear distinct. Likelihood-based pruning is a more direct countermeasure to mode-averaging by filtering out incompatible local modes. In contrast, iterative resampling seems to primarily address a different, albeit related, issue: global incoherence. This problem could arise even in compositional methods that do not use score averaging. The paper would be significantly strengthened by a more precise discussion on how iterative resampling specifically mitigates mode averaging, perhaps by explaining how improving global coherence helps resolve local mode conflicts.

- **Suggestions for Presentation and Clarity**: The paper's presentation could be improved in several areas.
	- **Clarity in TAMP Section**: The discussion on Task and Motion Planning (TAMP) in the main paper is brief and could be confusing for readers. Clarifying why it is a “hybrid-planning problem”, defining key terms like PDDL, and explaining the different representations of states/actions ($s_i$,$a_i$​) at high vs. low levels would be beneficial . Crucially, the fact that the GSC baseline is conditioned on an oracle task plan—a key factor for performance analysis—is only mentioned in Appendix D and should be in the main paper .
	- **Minor Errors and Formatting**: Several minor issues were noted.
	    - Line 316: The index in the set definition appears to have a typo ($s_h,a_h$ → $s_i,a_i$).
	    - Appendix G's title, "more details on robotic planning experiments," is misleading as it only covers TAMP, not OGBench.
	    - Some qualitative results could be discussed as limitations. For example, in the Appendix B visualization for the prompt “Silhouette wallpaper of a dreamy scene with shooting stars,” some global incoherence is still visible even applying iterative resampling and pruning.
	    - Minor capitalization and spacing issues were found in Appendices G.2 lines 1248, 1251, 1254 and H.1 between the paragraph and table 8 caption.
- **Compliance with LLM Usage Policy**: The authors do not state whether Large Language Models (LLMs) were used in preparing the manuscript, which is required by the ICLR 2026 Author Guide.

[1] Yoon, Jaesik, et al. "Monte carlo tree diffusion for system 2 planning." _arXiv preprint arXiv:2502.07202_ (2025).

[2] Yoon, Jaesik, et al. "Fast Monte Carlo Tree Diffusion: 100x Speedup via Parallel Sparse Planning." _arXiv preprint arXiv:2506.09498_ (2025).

[3] Lee, Gyubin, et al. "Adaptive Cyclic Diffusion for Inference Scaling." _arXiv preprint arXiv:2505.14036_ (2025).

**Questions:**

- In the 1D DDIM Inversion study (Appendix E, Fig. 8), the denoising latent paths (blue) appear to converge more strongly to the in-distribution modes compared to the DDIM inversion paths (red). Could you elaborate on the intuition or mechanism behind this visual result?
- The model utilizes a Mixture-of-Experts (MoE) architecture (Appendix G.1). What was the motivation for this specific choice over other potential architectures, and were any ablation studies performed to validate this design?
- The sampling strategy (Appendix G.2, Table 6) employs an adaptive schedule for resampling ($U(t)$) and specific ratios for exploration ($k_e$​) and pruning ($k_p$​). Have you experimented with simpler schedules (e.g., uniform resampling) or different values for $k_e$​ and $k_p$​ to test the sensitivity of these crucial hyperparameters?
- Appendix H.1 presents an ablation, CDGS (BFS-2), which uses system dynamics rollouts and is described as an “upper bound” of the proposed approach. Could you clarify what this result implies?

---

> ### Author Response · Authors · 2025-11-19
> **Rebuttal by Authors**
>
> Thank you for your valuable feedback. Please find our explanations below.
>
> **W1.1 Fixed Computational Overhead**
>
> The fixed computational overhead is because of batching. With no batching, the compute will increase for more complex tasks. In actual implementation, for a batch of $B$ candidate global plan for horizon $H$ each with $M$ local segments, we construct a batch of local segments of size $B\times M$ to denoise all the local segments in parallel for every denoising step. This step depends on the available GPU memory which limits the maximum batch size. We have provided more details on the compute scaling in Appendix M.
>
> **W1.2 Absence of Backtracking**
>
> We agree that backtracking is essential in domains where inference requires search over latent reasoning steps. However, that is beyond the scope of CDGS. Our setting is not a tree-search problem but a compositional consistency problem: the primary challenge is resolving local mode disagreements across overlapping trajectory factors In compositional diffusion, the global distribution is not explicitly represented, so backtracking over ``decision points" is not well-defined; what matters is whether a candidate global trajectory is locally self-consistent under the factorization. Extending to tree search is possible but challenging and an open research problem. This will be an interesting future direction.
>
> **W2 Ambiguous Link Between Iterative Resampling and Mode-Averaging**
>
> Our inference-time scaling method is based on the following idea: (1) composing feasible local plans leads to coherent global plan and (2) a coherent global plan will have feasible local plans. We observed that, due to mode-disagreements over shared coordinates during compositional sampling,  composing feasible local plans does not always lead to coherent global plan. We hypothesize that this is due to independently denoising local segments. To address this problem, CDGS uses iterative resampling to alternate compositional denoising and noising, allowing each local segment to repeatedly re-evaluate its prediction conditioned on its neighbors. This process propagates information across the entire chain, suppressing locally inconsistent mode combinations and reinforcing globally coherent ones.
>
> Once this reduces mode-disagreements (resampling cannot eradicate mode-averaging by itself, as shown in toy domain in Appendix M) and proposes better global plan candidates, pruning can filter out the remaining incoherent global plans.
>
> **W3.1 Clarity in TAMP Section**
>
> Thanks for pointing this out. We have updated the main paper with all the details. In summary, we have given a detailed explanation on why these problems are considered hybrid-planning problems and what is PDDL. We would further want to clarify that the GSC baseline we show in Table 2 does not use oracle task plan since we do not have diffusion experts per skill. In our case, GSC represents naive compositional diffusion. Appendix D shows how the availability of diffusion experts per skill and an oracle task plan will simplify the distribution from which CDGS is sampling without any privileged information.
>
> **W3.2 and W4** We have fixed the minor comments and added LLM usage statement. Thanks for pointing it out.
>
> **Q1. DDIM Inversion study**
>
> We apologize for the confusion. We have added a detailed explanation of the DDIM-inversion study in Appendix E.1 to clearly establish the importance of the pruning metric. In summary, DDIM inversion does forward noising to convert clean samples into gaussian and then we use the final noisy samples to do DDIM to retrieve the clean samples. The two paths do not mean the same thing.
>
> **Q2. MOE architecture**
>
> The MoE architecture of the score function is not critical to the contributions of CDGS. CDGS is a training-free framework that leverages pre-trained short-horizon/local diffusion models. We employed the MoE setup  particularly for the robotics experiment (since it is based on our custom dataset, so no pre-trained models were available) with the hypothesis that more experts will lead to better capturing of the multi-modal skill-transition structure of the dataset. We did not do any ablations.

---

> > ### Author Response · Authors · 2025-11-19
> > **Rebuttal continued**
> >
> > **Q3. Have you experimented with simpler schedules (e.g., uniform resampling) or different values for $k_p$ and $k_e$ to test the sensitivity of these crucial hyperparameters?**
> >
> > In most ideal scenario or simpler cases like the toy domain or the robotics setup, CDGS does not require any schedules. We can perform a fixed number of resampling iterations and a pruning step at every denoising step. However, this is limiting for larger local models like StableDiffusion2.1 and CogVideoX-1.5B. In that case, we need a schedule. Based on our analysis: (1) gradually increasing number of resampling steps throughout the denoising process (until specified U is achieved, then constant) and (2) use pruning for last 20% of the denoising process. This is sufficient to reliably sample a feasible global plan and also reduces computational requirements. We provide more details in Appendix M.
> >
> > **Q4. Appendix H.1 presents an ablation, CDGS (BFS-2)**
> >
> > We perform this analysis to use CDGS as a learned prior for performing BFS for TAMP. The complete pipeline is: (1) for a given current state, sample global plans and group them based on the first skill in the plans. (2) take the top-2 skills with majority voting and use physics simulator to find feasible transitions and then repeat the process. This allows us to constraint BFS tree to a branching factor of 2 and leverages simulator for proceeding in the environment. Both the tree search and simulator are priviledged information not available to CDGS and hence CDGS (BFS-2) is an upper bound.
> >
> > Thanks again. We will be happy to answer any other follow up questions.

---

> > > ### Comment · Reviewer_Qy6u · 2025-11-24
> > >
> > > Thank you for the detailed response and additional experiments. Regarding W2, I still find the link between iterative resampling and mode-averaging tenuous, as you also acknowledged (`resampling cannot eradicate mode-averaging by itself`).
> > >
> > > To address this, I suggest broadening the problem scope in the Introduction. Explicitly framing "global incoherence" (or the inability to generate globally plausible candidates) as a key limitation of prior methods (it is clearly shown in your new experiment comparing resampling=1 vs resampling=80), alongside mode-averaging would provide a solid logical justification for iterative resampling and significantly strengthen the paper's narrative.

---

### Official Review · Reviewer_QzLm · 2025-10-30

**Soundness:** 2
**Presentation:** 3
**Contribution:** 2
**Rating:** 6
**Confidence:** 5

**Summary:**

The paper proposes a novel method, Compositional Diffusion with Guided Search (CDGS), for long-horizon planning tasks. The core idea is to combine the generative power of diffusion models with a structured search algorithm to create complex and coherent plans over long horizons. CDGS works by composing shorter-horizon plans generated by a diffusion model and then using an iterative resampling and pruning mechanism to guide the search towards feasible and high-quality long-horizon plans. The authors demonstrate the effectiveness of their approach on a variety of challenging tasks, including robotic manipulation, long-form video generation, and panoramic image synthesis, showing significant improvements over existing methods.

**Strengths:**

1. The paper introduces a highly novel and significant approach to long-horizon planning. The combination of compositional generative models (diffusion models) with explicit guided search is a powerful paradigm that effectively balances generation diversity with goal-directedness. This work has the potential to influence future research in planning, robotics, and generative modeling.
2. The experimental evaluation is thorough and convincing. The authors demonstrate the superiority of CDGS across multiple, distinct, and challenging domains. The qualitative results, especially in video and panorama generation, are impressive and clearly showcase the model's ability to maintain long-term coherence. The quantitative results on robotic planning tasks also show clear improvements over state-of-the-art baselines.
3. The proposed method is well-grounded in mathematical principles. The use of compositional score functions and the DDIM-based pruning objective are elegant and well-motivated. The paper provides sufficient detail to understand the core mechanics of the algorithm, and the mathematical derivations appear to be correct and consistently applied.

**Weaknesses:**

1. The proposed method seems computationally intensive. The iterative nature of resampling and refining plans at each step of the search could be very time-consuming, which might limit its applicability in real-time or resource-constrained settings. A more detailed analysis of the computational complexity and wall-clock time comparisons with baselines would be beneficial.
2. The paper could benefit from providing more details on how the states and actions for the robotic planning tasks are represented and fed into the diffusion model. Understanding the specifics of the parameterization would help in assessing the method's applicability to other robotic tasks.
3. While the paper does present some ablation studies (e.g., in Appendix H), a more in-depth analysis of the contribution of each component of CDGS would strengthen the paper. For instance, a clearer study on the comparison of different pruning objectives could provide valuable insights.

**Questions:**

1. The pruning objective $g(x_0^{i})$ is based on the curvature of the denoising path. While intuitive, could the authors provide a more formal justification for why this particular metric is a good proxy for plan feasibility or quality? Have the authors experimented with other pruning criteria, and how sensitive is the performance of CDGS to the choice of this objective?
2. How does the performance and computational cost of CDGS scale as the planning horizon $H$ increases? Is there a point where the composition of many short-horizon plans leads to a degradation in quality or an explosion in computational requirements?
3. The paper mentions the use of a Mixture-of-Experts (MoE) layer in the score model. How critical is the MoE component for the success of CDGS, especially in multi-modal tasks?
4. The results presented are very positive. It would be instructive to see some failure cases of CDGS. In which scenarios does the method struggle to produce coherent long-horizon plans? This could shed light on the limitations of the current approach and point to avenues for future work.

---

> ### Author Response · Authors · 2025-11-19
> **Rebuttal by Authors**
>
> Thank you for your valuable feedback. Please find our explanations below.
>
> **W1. Runtime analysis**
>
> We have provided a detailed runtime analysis in the Appendix M, please see the revised paper. In summary, CDGS incurs a predictable linear overhead relative to na\"ive compositional diffusion, and in practice the additional cost scales up the performance significantly as well (the core essence of inference-time scaling). Across domains, we observe that CDGS is $(0.5U + 0.1T)$x more expensive to run where $U$ is the number of resampling steps and $T$ is the number of denoising iterations. In most of our experiments, we conclude that CDGS typically requires 10–12x more compute than a baseline compositional sampler when both resampling and pruning are enabled (across OGBench and panorama synthesis). The exact wall clock time depends on the base local generative model and the number of inference steps required to generate a good sample from it. Engineering optimizations—batching local segments, limiting pruning to low-noise steps, and JIT compilation—keep the wall-clock scaling low. Importantly, this computational cost consistently yields substantial improvements in global coherence across all evaluated domains.
>
> It is worth noting that because of independently sampling local segments, the complexity of the problem increases exponentially with horizon. For a horizon of $H$ and each local segment having $Z$ feasible modes, there can be $Z^H$ possible sequences of feasible global plan modes, only a few of them will be valid for coherent global plan synthesis. CDGS is able to navigate this exponentially increasing domain by linearly scaling the compute and memory requirements.
>
> **W2. more details on how the states and actions for the robotic planning tasks are represented and fed into the diffusion model.**
>
> We provided the details in Appendix F.2. In summary, CDGS uses low-dim state representation for the robotics setup where the state vector is formed by concatenating 6D poses of the objects in the scene. For TAMP benchmark, we use skill-based action representation as provided in Appendix F.1 and, for OGbench tasks, we use end-effector relative motion as the action following their Github codebase. Appendix G shows how we feed the state-skill-action vectors into the score model.
>
> **W3. clearer study on the comparison of different pruning objectives could provide valuable insights.**
>
> We have provided an empirical comparison of the different pruning methods in Appendix E.1. In summary we see two directions of improvement when using CDGS's curvature based metric vs other reconstruction-based alternatives:
> - DDIM inversion only requires forward noising while reconstruction methods require both forward noising and denoising back (for multiple choices of diffusion timestep).
> - For distributions with disjoint modes (like the one considered for our 1D experiment), it is not necessary that the reconstructed sample after noising and denoising will belong to the same mode as the original clean sample. This makes reconstruction based metrics invalid or overly conservative neglecting in-distribution segments.
>
> **Q1. Validation of the pruning metric**
>
> Although a fully rigorous theoretical equivalence between the time-derivative of the score function and likelihood is not yet available in diffusion theory, recent papers have started looking at this direction for OOD detection. While [1] uses powers of curvature as features to identify OOD, [2] (a concurrent work) realized that arbitrary movements happens only near low-noise regions. The relationship between DDIM-inversion curvature and likelihood has a clear geometric basis that motivates our pruning criterion, and we empirically validate it on controlled toy settings. Under DDIM inversion, a candidate sample $x_0$ evolves according to a probability-flow–like ODE whose vector field depends on the learned score function. If $x_0$ lies far from the true data manifold, the score-function applies high magnitude scores (velocities) in abrupt directions (as it has never seen the sample while training) to steer the inversion trajectory towards high-density region.
>
> In contrast, in-distribution samples follow latent data manifolds, producing smooth, slowly varying score predictions. Thus, the cumulative ``curvature" is an indicator of how much the model must change a candidate to make it consistent with the learned distribution. We empirically validate this intuition in a 1D setting as shown Appendix E.1.
>
> While SVGD based heuristics do not directly apply for the compositional setting, a future version of SVG-MPPI for compositional generative sampling can leverage global coherency metrics like our pruning metric.
>
> [1] Heng, Alvin, and Harold Soh. "Out-of-distribution detection with a single unconditional diffusion model." (2024)
> [2] Barkley, Brett, Preston Culbertson, and David Fridovich-Keil. "SCOPED: Score-Curvature Out-of-distribution Proximity Evaluator for Diffusion." (2025).

---

> > ### Author Response · Authors · 2025-11-19
> > **Rebuttal continued**
> >
> > **Q2. How does the performance and computational cost of CDGS scale as the planning horizon $H$
> >  increases?**
> >
> > We have provided all the details in Appendix M. CDGS scales linearly with planning horizon in terms of computational cost. In terms of performance, with fixed resampling steps, the performance drops (almost) exponentially with horizon. In terms of memory requirements, in practical implementation, for a batch of $B$ candidate global plans for horizon $H$ each with $M$ local segments, we construct a batch of local segments of size $B\times M$ to denoise all the local segments in parallel for every denoising step. This step depends on the available GPU memory which limits the maximum batch size.
> >
> > **Q3. How critical is the MoE component for the success of CDGS, especially in multi-modal tasks?**
> >
> > The MoE architecture of the score function is not critical to the contributions of CDGS. CDGS is a training-free framework that leverages pre-trained short-horizon/local diffusion models. We employed the MoE setup  particularly for the robotics experiment with the hypothesis that more experts will lead to better capturing of the multi-modal skill-transition structure of the dataset.
> >
> > **Q4. limitations of the current approach**
> >
> > CDGS is a add-on to any pre-train diffusion model sampling algorithm that scales the sampling capabilities to horizons beyond training data. Hence the performance of CDGS is heavily dependent on the quality of the base short-horizon diffusion model. Some additional limitations of CDGS include:
> > - Assume given goal condition: Similar to prior works like GSC and CompDiffuser, CDGS plans by generating a long-horizon plan from the start state to a given goal state. It would be interesting to see how our method could be integrated with classifier guidance for goal objectives or goal-generation methods.
> > - Fixed plan horizon: In this work, we focus on generating a global plan for a given planning horizon, and we leave the problem of selecting the appropriate planning horizon to future work. Intuitively, CDGS can sample a suitable solution for any specified plan horizon given the same start and goal, which allows us to find the best among multiple choices for $H$.
> >
> > We will add this to the limitation section.
> >
> >
> >
> > Thanks again. We will be happy to answer any other follow up questions.

---

> > > ### Comment · Reviewer_QzLm · 2025-11-26
> > >
> > > I appreciate the authors thorough response to all my concerns with additional experiments. These have all been addressed, and I have no further concerns, I have increased my score to 8 accordingly.

---

### Official Review · Reviewer_27Ji · 2025-10-31

**Soundness:** 2
**Presentation:** 3
**Contribution:** 3
**Rating:** 6
**Confidence:** 5

**Summary:**

The paper presents Compositional Diffusion with Guided Search (CDGS), a generative method that augments compositional diffusion sampling with population-based guided search and likelihood-based pruning to mitigate mode-averaging when composing multimodal local distributions. The proposed method aims to unify diffusion-based planning and long-horizon content generation under a single approach, demonstrating its effectiveness in not only robotic long-horizon manipulation tasks but also in panoramic imaging and video generation. While the motivation is compelling and the experiments effectively highlight the contributions, the work overemphasizes conceptual novelty and lacks rigorous comparisons with modern and traditional planning-based methods.

**Strengths:**

1. The paper identifies a genuine issue in compositional generative modeling: mode-averaging across multimodal local distributions, which degrades global coherence in long-horizon tasks. In addition, CDGS is evaluated across robotics, image synthesis, and video generation, demonstrating domain-agnostic adaptability at the conceptual level.

2. The method is implemented end-to-end with population-based inference and ablations (with/without pruning, resampling, scaling analysis), demonstrating the engineering completeness. Despite limited novelty, the approach achieves reasonable performance improvements on OGBench, TAMP, and long-horizon generative benchmarks.

3. The figures, especially Figure 4, effectively illustrate the proposed resampling and pruning mechanisms in an end-to-end manner. Algorithms and Equations are clearly presented, with a strong emphasis on reproducibility.

**Weaknesses:**

1. The population-based denoising, iterative resampling, and pruning loops likely incur heavy compute costs. No runtime or complexity analysis is provided to justify practicality. Classical planners, such as SVG-MPPI [1] and Reverse-KL MPPI [2], already produce mode-seeking behavior in closed form, with stronger theoretical rigor and lower computational overhead. Within this realm, CDGS lacks a clear justification for its adoption over these established alternatives.

2. CDGS essentially reuses established ideas, including population-based search, iterative resampling, and likelihood pruning. Indeed, it repackages under the umbrella of diffusion inference without introducing fundamentally new theory or algorithmic principles. The paper lacks direct traditional predecessors such as SVG-MPPI [1], Reverse-MPPI [2], and SV-MPC [3], which already address multimodality via reverse KL minimization and Stein variational guidance. Without such baselines, it is unclear how CDGS advances beyond established mode-seeking control frameworks. More importantly, the paper fails to position itself in relation to modern diffusion-based mode-seeking methods, such as Flow to the Mode [4] and Mean-Shift Distillation [5], which explicitly address mode collapse and mode-averaging in generative diffusion models.

3. The pruning metric J(τ) based on DDIM curvature is heuristic and unvalidated. There is no proof linking it to likelihood, feasibility, or global convergence. In contrast, SVG-MPPI provides a clear variational derivation and theoretical backing. Moreover, the stability and safe validation are largely omitted, which might be acceptable for video generation or image synthesis, but not for robot control for long-horizon tasks.

4. The experiments presented are too limited in scope to establish the superiority of CDGS convincingly. The evaluation mainly contrasts against lightweight or outdated baselines (e.g., GSC, Diffuser, CEM-based methods) and omits stronger diffusion-based or traditional mode-seeking approaches, such as Flow to the Mode [4], Mean-Shift Distillation [5], and SVG-MPPI [1]. The claims of generality and cross-domain performance are not substantiated by statistically rigorous analysis or comprehensive comparisons. Without large-scale quantitative studies, ablations on computational trade-offs, or benchmarks against state-of-the-art mode-seeking diffusion and planning frameworks, the empirical validation remains insufficient to justify the claimed novelty or practical advantage of the proposed method.

[1] Honda, Kohei, et al. "Stein variational guided model predictive path integral control: Proposal and experiments with fast maneuvering vehicles." 2024 IEEE International Conference on Robotics and Automation (ICRA). IEEE, 2024.
[2] Kobayashi, Taisuke, and Kota Fukumoto. "Real-time sampling-based model predictive control based on reverse kullback-leibler divergence and its adaptive acceleration." arXiv preprint arXiv:2212.04298 (2022).
[3] Lambert, Alexander, et al. "Stein variational model predictive control." arXiv preprint arXiv:2011.07641 (2020).
[4] Sargent, Kyle, et al. "Flow to the mode: Mode-seeking diffusion autoencoders for state-of-the-art image tokenization." arXiv preprint arXiv:2503.11056 (2025).
[5] Thamizharasan, Vikas, et al. "Mean-Shift Distillation for Diffusion Mode Seeking." arXiv preprint arXiv:2502.15989 (2025).

**Questions:**

1. Could you position CDGS against modern mode-seeking control/planning frameworks (e.g., Reverse-KL MPPI / SV-MPC / SVG-MPPI) and mode-seeking diffusion methods (e.g., mean-shift/flow-to-mode)? Why is CDGS preferable when the objective is to avoid “mode averaging”?
2. Why is multiplicative aggregation over segments (Eq. 5) preferable to additive/soft-min aggregations? Any sensitivity analysis to the aggregation choice?
3. Figure 5 shows the benefits from increasing $B$ and $U$. Where are the diminishing returns, and how do you set $\lambda_t$ (exploration/exploitation) across timesteps? Provide a schedule and an ablation.
4. In OGBench (Table 1), please report the number of seeds, confidence intervals, and effect sizes vs. the strongest baseline (not only means). Also, clarify whether CDGS uses less long-horizon supervision than IRL baselines in practice.
5. Do you enforce hard preconditions/effects (e.g., in-hand (hook)) during sampling to guarantee task-level feasibility, or is feasibility entirely delegated to DDIM-curvature pruning?

---

> ### Author Response · Authors · 2025-11-19
> **Rebuttal by Authors**
>
> Thank you for your valuable feedback. Please find our explanations below.
>
> **W1. Position of the paper and comparison with SVG-MPPI, Flow to Mode, etc baselines**
>
> Thanks for the insightful comments. We agree that SVG-MPPI is an inspiring and relevant work, but it focuses on an adjacent (yet fundamentally different) problem: it seeks a single mode of a *monolithic* global plan distribution, whereas CDGS addresses mode selection and consistency in the *compositional* diffusion setting, where no global distribution is available.
>
> SVG-MPPI assumes a single global generative model over full-horizon plans; inference operates directly within this joint distribution. CDGS, in contrast, only has access to overlapping local plan distributions and must construct a coherent global plan at test time by composing these local factors. This structural difference leads to distinct inference challenges and limits the appropriateness of direct comparison.
>
> - **Different sources of mode-seeking difficulty.** In monolithic models, mode-seeking concerns the multimodality of a single joint distribution: inference may average across globally distinct modes. In compositional diffusion, multimodality arises at the level of each local factor. The challenge is to align locally chosen modes so that their shared coordinates agree. Mode-averaging here stems directly from the compositional structure—e.g., selecting incompatible high-probability modes across factors—which has no analogue in monolithic models.
> - **Applying SVG-MPPI-style methods to compositional sampling is an interesting open problem.**  SVG-MPPI assumes access to a single global optimal action distribution over the entire horizon and uses SVGD to pull guide particles toward one mode of that global distribution before performing a closed-form MPPI update. In the compositional setting, however, there is no global distribution to evaluate SVGD gradients against; instead, only overlapping local factors are available, each with its own multimodal structure. Applying SVG-MPPI would require constructing a surrogate “global” target distribution on the fly from exponentially many combinations of local modes, designing a Stein-type update that simultaneously moves particles to consistent modes across all factors, and defining an MPPI update that enforces agreement on shared coordinates. We consider this as a nontrivial open research problem.
>
> **W2. Runtime analysis**
>
> We have provided a detailed runtime analysis in the Appendix M, please see the revised paper. In summary, CDGS incurs a predictable linear overhead relative to na\"ive compositional diffusion, and in practice the additional cost scales up the performance significantly as well (the core essence of inference-time scaling). Across domains, we observe that CDGS is $(0.5U + 0.1T)$x more expensive to run where $U$ is the number of resampling steps and $T$ is the number of denoising iterations. In most of our experiments, we conclude that CDGS typically requires 10–12x more compute than a baseline compositional sampler when both resampling and pruning are enabled (across OGBench and panorama synthesis). The exact wall clock time depends on the base local generative model and the number of inference steps required to generate a good sample from it. Engineering optimizations—batching local segments, limiting pruning to low-noise steps, and JIT compilation—keep the wall-clock scaling low. Importantly, this computational cost consistently yields substantial improvements in global coherence across all evaluated domains.
>
> It is worth noting that because of independently sampling local segments, the complexity of the problem increases exponentially with horizon. For a horizon of $H$ and each local segment having $Z$ feasible modes, there can be $Z^H$ possible sequences of feasible global plan modes, only a few of them will be valid for coherent global plan synthesis. CDGS is able to navigate this exponentially increasing domain by linearly scaling the compute and memory requirements.

---

> ### Author Response · Authors · 2025-11-19
> **Rebuttal continued**
>
> **W3. Validation of the pruning metric**
>
> Although a fully rigorous theoretical equivalence between the time-derivative of the score function and likelihood is not yet available in diffusion theory, recent papers have started looking at this direction for OOD detection. While [1] uses powers of curvature as features to identify OOD, [2] (a concurrent work) realized that arbitrary movements happens only near low-noise regions. The relationship between DDIM-inversion curvature and likelihood has a clear geometric basis that motivates our pruning criterion, and we empirically validate it on controlled toy settings. Under DDIM inversion, a candidate sample $x_0$ evolves according to a probability-flow–like ODE whose vector field depends on the learned score function. If $x_0$ lies far from the true data manifold, the score-function applies high magnitude scores (velocities) in abrupt directions (as it has never seen the sample while training) to steer the inversion trajectory towards high-density region.
>
> In contrast, in-distribution samples follow latent data manifolds, producing smooth, slowly varying score predictions. Thus, the cumulative ``curvature" is an indicator of how much the model must change a candidate to make it consistent with the learned distribution. We empirically validate this intuition in a 1D setting as shown Appendix E.1.
>
> While SVGD based heuristics do not directly apply for the compositional setting, a future version of SVG-MPPI for compositional generative sampling can leverage global coherency metrics like our pruning metric.
>
> [1] Heng, Alvin, and Harold Soh. "Out-of-distribution detection with a single unconditional diffusion model." Advances in Neural Information Processing Systems 37 (2024)
>
> [2] Barkley, Brett, Preston Culbertson, and David Fridovich-Keil. "SCOPED: Score-Curvature Out-of-distribution Proximity Evaluator for Diffusion." arXiv preprint arXiv:2510.01456 (2025).
>
> **W4. Scope of experiments**
>
> We agree that strong empirical validation is important; however, the baselines suggested by the reviewer—Flow-to-Mode, Mean-Shift Distillation, SVG-MPPI—operate in a different modeling regime. These methods assume a single global trajectory distribution and perform monolithic mode-seeking within that model, whereas our method focuses on composing multiple generative models.
>
> The baselines we compare against—GSC, Diffuser, CEM-based compositional methods—are all specifically designed for compositional sampling. As we have discussed in the main text (Lines 350-403), they are either limited by manual specifications (like PDDL), lack of geometrical understanding (like LLM/VLM) or the mode-averaging (GSC) and infeasibility issues that CDGS is designed to fix. Across tasks, CDGS consistently recovers coherent global plans where these methods fail, validating our core contribution: an inference-time mechanism that makes compositional generative modeling reliable for sequential decision-making.
>
> We agree that broader large-scale evaluation is valuable, but we emphasize that the present experiments already demonstrate the generality and effectiveness of CDGS across *three distinctive domains*—image generation, video prediction, and decision-making. These settings differ significantly in data modality, temporal structure, and action/state semantics, yet CDGS yields consistent improvements over compositional baselines in all of them.
>
> **Q1. position CDGS against modern mode-seeking control/planning frameworks ...**
>
> Thank you for the question. This gave us an opportunity to clarify what *mode-averaging* means for compositional sampling and how it is different from the *mode-seeking* monolithic generative sampling literature. Please find our detailed explanation above.
>
> **Q2. multiplicative aggregation over segments ...**
>
> Equation 5 in the main paper defines the verifier objective by assigning each local plan a curvature based coherence score and aggregating these scores multiplicatively across the overlapping factors. This product-of-exponentials matches the factor-graph view of compositional diffusion: each window contributes an independent consistency term, and $\log J$ becomes a sum of local energies. Importantly, the specific choice of aggregation is not critical to CDGS; its role is simply to sharply penalize trajectories containing any locally inconsistent segment while preserving trajectories whose local plans are mutually coherent. Alternative monotone aggregations could be used, but they would serve the same functional purpose within our inference-time framework.

---

> > ### Author Response · Authors · 2025-11-19
> > **Rebuttal continued**
> >
> > **Q3. Where are the diminishing returns, schedule and ablations**
> >
> > We conduct additional experiments for longer horizon planning in the illustrative 1-D domain, as shown in Appendix M. We observe that with increasing horizon, we do not see much improvement with scaling number of resampling steps. Since the pruning performance depends a lot on the proposer's capability to sample good candidates, the pruning performance drops too.
> >
> > Based on our analysis: (1) gradually increasing number of resampling steps throughout the denoising process (until specified U is achieved, then constant) and (2) use pruning for last 20\% of the denoising process. This is sufficient to reliably sample a feasible global plan and also reduces computational requirements.
> >
> > **Q4. OGBench (Table 1), please report the number of seeds, confidence intervals ...**
> >
> > CDGS uses short-segments of 80 frames that correspond to 4 secs of motion plan at 20Hz. We empirically found that this is sufficient to represent atomic skills like open/close drawer/window door and press buttons. Further, the results are reported for 5 seeds (thanks for pointing this out). We have updated the table.
> >
> > **Q5. Do you enforce hard preconditions/effects ...**
> >
> > We do not enforce hard preconditions/effects during sampling. Because each skill affects only a small, skill-specific subset of objects, we use a learned forward dynamics model for each skill to ensure that predicted local transitions obey this constraint. Concretely, for every planned local transition in $\{(s_i, \pi_i, a_i, s_{i+1})\}$, we overwrite the next state using the corresponding skill forward dynamics network $s_{i+1} = f_{\pi_i}(s_i, a_i)$, which updates only the objects relevant to the predicted skill $\pi_i$ and keeps all other objects static. This produces locally valid transitions consistent with the TAMP skill semantics, allowing DDIM-curvature pruning to evaluate feasibility accurately. More details and the integration of this step into the CDGS pipeline is shown in Algorithm N of the appendix.
> >
> >
> > Thanks again. We will be happy to answer any other follow up questions.

---

> ### Comment · Reviewer_27Ji · 2025-11-26
>
> I appreciate the response of the authors, which answers all my concerns. The paper deserves a solid acceptance.
> Therefore, I raise the score to 8.

---

### Meta-Review · Area_Chair_ArF2 · 2025-12-17

**Summary:**

The paper received initial positive reviews with scores of 6, 6, 8, and 6. The area chair reviewed both the feedback and the authors' rebuttal and confirmed that most of the reviewers' concerns were adequately addressed. The final average score is expected to be 8. Consequently, the area chair recommends accepting the paper.

**Reviewer Concerns:**

**Reviewer 27Ji**, **Reviewer QzLm**, and **Reviewer xJxE** all agree that the authors' responses have sufficiently addressed their concerns. The area chair supports the reviewers' opinions.

**Reviewer Qy6u** shows a remaining concern about "the ambiguous link between iterative resampling and mode-averaging" and thus provides additional suggestions for broadening the problem scope in the Introduction. It is suggested that
> explicitly framing "global incoherence" (or the inability to generate globally plausible candidates) as a key limitation of prior methods, alongside mode-averaging, would provide a solid logical justification for iterative resampling and significantly strengthen the paper's narrative.

Nevertheless, the reviewer's suggestions do not seem to affect the final decision.

**Reviewer Scores:**

The AC thinks that **Reviewer 27Ji**, **Reviewer QzLm**, and **Reviewer xJxE** will increase their scores to 8 according to their responses:
> Reviewer 27Ji: I appreciate the response of the authors, which answers all my concerns. The paper deserves a solid acceptance. Therefore, I raise the score to 8.

> Reviewer QzLm: I appreciate the authors' thorough response to all my concerns with additional experiments. These have all been addressed, and I have no further concerns. I have increased my score to 8 accordingly.

> Reviewer xJxE: I thank the authors for the detailed clarifications. All of my concerns have been well addressed, and I raised my score accordingly. I enjoyed reading this work. Good luck!

The initial review of **Reviewer Qy6u** is positive. The follow-up response from **Reviewer Qy6u** acknowledges the authors' rebuttal and raises no further concerns. The AC assumes the reviewer will maintain the score as 8.

---

The final scores are expected to be **8**, **8**, **8**, and **8**.

---

### Decision · Program_Chairs · 2026-01-26

Accept (Oral)